# An Information-Theoretic Framework for Understanding Learning and Choice Under Uncertainty

**DOI:** 10.3390/e27101056

**Published:** 2025-10-11

**Authors:** Jae Hyung Woo, Lakshana Balaji, Alireza Soltani

**Affiliations:** 1Department of Psychological and Brain Sciences, Dartmouth College, Hanover, NH 03755, USA; jae.hyung.woo.gr@dartmouth.edu; 2Department of Biology, Indian Institute of Science Education and Research Tirupati (IISER T), Tirupati 517619, Andhra Pradesh, India; lakshanabalaji@students.iisertirupati.ac.in

**Keywords:** reinforcement learning, value-based decision making, conditional entropy, mutual information, uncertainty

## Abstract

Although information theory is widely used in neuroscience, its application has primarily been limited to the analysis of neural activity, with much less emphasis on behavioral data. This is despite the fact that the discrete nature of behavioral variables in many experimental settings—such as choice and reward outcomes—makes them particularly well-suited to information-theoretic analysis. In this study, we provide a framework for how behavioral metrics based on conditional entropy and mutual information can be used to infer an agent’s decision-making and learning strategies under uncertainty. Using simulated reinforcement-learning models as ground truth, we illustrate how information-theoretic metrics can reveal the underlying learning and choice mechanisms. Specifically, we show that these metrics can uncover (1) a positivity bias, reflected in higher learning rates for rewarded compared to unrewarded outcomes; (2) gradual, history-dependent changes in the learning rates indicative of metaplasticity; (3) adjustments in choice strategies driven by reward harvest rate; and (4) the presence of alternative learning strategies and their interaction. Overall, our study highlights how information theory can leverage the discrete, trial-by-trial structure of many cognitive tasks, with the added advantage of being parameter-free as opposed to more traditional methods such as logistic regression. Information theory thus offers a versatile framework for investigating neural and computational mechanisms of learning and choice under uncertainty—with potential for further extension.

## 1. Introduction

Information theory has been used widely across different domains of cognitive and systems neuroscience, ranging from the analysis of spike trains to estimation of uncertainty in sensory environments. For example, Shannon entropy and related metrics have been used to study information flow [1,2,3,4], functional and effective connectivity [5,6,7,8,9], and variability in neural response [10,11,12,13,14]. These quantitative approaches have provided significant insight into brain functions at the cellular and system levels.

Despite this widespread use of information-theoretic metrics in analyzing neural data, their application in investigating behavior has been surprisingly limited. Although several studies have employed entropy-based metrics to quantify uncertainty in stimuli [15,16,17,18] and outcomes [19,20,21], only a few studies have utilized information theory to directly examine the underlying decision-making and learning mechanisms [22,23,24]. This is despite the fact that many behavioral measures—such as binary choices and reward feedback—are ideally structured for analysis using information-theoretic tools.

A common tool for studying learning and choice behavior is reinforcement learning (RL) models [25,26,27,28,29,30], due to their simplicity and interpretability. In this approach, various RL models are constructed and fit to the choice data. The best-fitting model is then identified through model selection, and its components form the basis for inferences about underlying cognitive and/or neural mechanisms. However, these models often need to be augmented with additional components to capture empirical data in specific tasks (e.g., [31,32,33,34]), while it remains unclear how many components are necessary—or sufficient—to define the “best” model. Such extensions include separate learning rates for positive versus negative prediction errors (differential learning rates), time-varying (dynamic) learning rates, modulations of decision-making by reward harvest rate, and arbitration between alternative models of the environment, among others [35,36,37,38,39,40,41,42]. Nevertheless, currently, there is no systematic method to identify the critical components required in RL models. This lack of methodology can lead to important mechanisms being overlooked, even in models that provide the best fit among the tested models.

Interestingly, choice behavior under uncertainty is inherently stochastic, making it naturally suited for analysis through information theory. Recently, a set of information-theoretic metrics was proposed as a model-agnostic approach to uncover the hidden learning mechanisms underlying behavior. For example, Trepka et al. [32] have suggested that behavioral metrics based on conditional entropy can quantify the consistency in local choice strategies and predict undermatching. Moreover, Woo et al. [43] have shown that mutual information, alongside other measures, can capture the influence of high-order statistics of the reward environment. These results highlight the potential of information-theoretic metrics to probe learning and decision-making processes beyond what traditional model-fitting techniques can reveal.

Here, we extended this approach by applying information-theoretic metrics to simulated choice data from different RL models—serving as ground truth—across four learning tasks. We show that this method captures key aspects of learning and decision making without relying on model fitting. To that end, we used existing metrics and developed new ones to detect a range of learning and decision-making mechanisms and their dynamic adjustments. We begin by identifying higher learning rates following positive prediction errors (rewarded outcomes) than following negative prediction errors (unrewarded outcomes)—a phenomenon referred to as positivity bias [44,45,46,47]. We also examine changes in learning rates over time as a result of metaplasticity [48,49]. Next, we investigate the influence of reward harvest rate, which has been shown to modulate learning and decision making [36,37,50,51]. Finally, in naturalistic reward environments, choice options often possess multiple attributes, each potentially predictive of reward outcomes. Previous studies have demonstrated that humans and other animals tackle this challenge by simultaneously learning about alternative reward contingencies (i.e., models of the environment), arbitrating between these models, and deploying selective attention to guide differential learning and decision making [40,52,53,54,55,56,57,58,59]. Therefore, we test how information-theoretic metrics can be used to detect the presence of distinct learning strategies in complex reward environments. Overall, our results demonstrate that the patterns of information-theoretic metrics provide useful summary statistics of the behavioral signatures generated by different learning and choice mechanisms, thereby offering a complementary approach to model fitting (both discovery and recovery).

## 2. Materials and Methods

### 2.1. General Experimental Paradigm: Probabilistic Reversal Learning

We simulated choice data using different reinforcement learning (RL) agents performing two variants of the probabilistic reversal learning task (PRL), a widely used paradigm to assess cognitive flexibility across various species [60]. In a typical probabilistic reversal learning paradigm, subjects choose between two options based on the reward feedback they receive on each trial, with the selection of each option associated with a different probability of reward. Choice options can take various forms, such as distinct visual stimuli on a computer screen, identical stimuli presented at different spatial locations, or physical levers. One option yields a reward with a higher probability (the better option) than the other (the worse option), but these reward contingencies switch at fixed or random times within an experimental session, creating ’reversals.’ Crucially, reversals are not signaled and thus, subjects have to adjust their choice solely based on reward feedback to maximize their chance of winning a reward. Commonly used rewards include drops of juice or water (e.g., [61,62]), sucrose pellets (e.g., [63]), and monetary incentives for humans (e.g., [58,64]), which are delivered probabilistically at the end of each trial. For most simulations, we set the reward probabilities at 80% and 20% (corresponding to an 80/20 reward schedule), with each block consisting of 80 trials, unless stated otherwise. Each block contains a reversal in the middle of the block, where the reward probabilities of the two options switch. In the final set of simulations (Section 3.4), we also considered a generalized PRL task in which reward probabilities depended on one feature of each choice option (e.g., its color or its shape), while the other feature carried no information about the reward.

### 2.2. Information-Theoretic Metrics

In this study, we utilized and extended the information-theoretic metrics introduced previously [32,43]. These measures—grounded in conditional entropy, mutual information, and outcome-specific decompositions—quantify how past rewards and choices shape the uncertainty surrounding an agent’s decision to stay with or switch from its previous choice (Figure 1b).

In general, for discrete random variables *X* and *Y*, the conditional entropy H(Y|X) represents the remaining uncertainty in Y given the information about the variable X. Formally, it is defined as(1)$$\begin{array}{cc} H(Y|X) & =\sum_{x\in X}P\left(x\right)\cdot H\left(Y|X=x\right)\\ \; & =-\sum_{x\in X}\sum_{y\in Y}P\left(x,y\right)\log_{2}\frac{P(x,y)}{P(x)} \end{array}$$

Lower values of conditional entropy indicate that knowing the values of *X* reduces the uncertainty in *Y*, suggesting a strong dependence between the two variables.

Similarly, mutual information, denoted as I(X;Y), quantifies the information shared between discrete variables *X* and *Y*. Higher values of I(X;Y) indicate a greater dependency between variables, such that knowledge of *Y* would make *X* more predictable (less uncertain). This relationship is expressed using the difference between Shannon entropy for *Y*, H(Y), and the conditional entropy, H(Y|X):(2)$$\begin{array}{cc} I(X;Y) & =H(Y)-H(Y|X)\\ \; & =\sum_{x\in X}\sum_{y\in Y}P\left(x,y\right)\log_{2}\frac{P(x,y)}{P(x)P(y)} \end{array}$$

Building on these general formulations, we define behavioral metrics to quantify uncertainty in the agent’s choice strategy, specifically in terms of whether the agent “stays” with or “switches” from the previous choice option, given certain outcomes (Figure 1a). Specifically, we aim to quantify how uncertainty in choice strategy is reduced by certain task-related information: the previous reward outcome (*R*) and the previously chosen option (Opt). The resulting metrics include the conditional entropy of reward-dependent strategy (ERDS), the conditional entropy of option-dependent strategy (EODS), and the conditional entropy of reward and option-dependent strategy (ERODS). These are paired with mutual information metrics including mutual information between reward outcome and choice strategy (MIRS), mutual information between previous choice option and choice strategy (MIOS), and mutual information between reward outcome, choice options, and choice strategy (MIROS).

More specifically, ERDS measures the influence of previous reward outcomes—reward vs. no reward (referred to as win vs. loss for simplicity)—on the uncertainty of the subsequent choice strategy (stay vs. switch), as follows:(3)$$\begin{array}{cc} \mathrm{ERDS} & =H(Str_{t}|R_{t-1})\\ \; & =H(Str)-I(R;Str)\\ \; & =-\sum_{R\in \{\mathrm{win},\mathrm{loss}\}}\sum_{Str\in \{\mathrm{stay},\mathrm{switch}\}}P\left(R,Str\right)\log_{2}\frac{P(R,Str)}{P(R)}. \end{array}$$
where $Str_{t}$ or Str denotes the choice strategy between two subsequent trials (stay = 1, switch = 0), $R_{t-1}$ is the reward outcome on the previous trial (reward or win =1 and no reward or loss =0), $P(R_{t-1},Str_{t})$ is the joint probability of choice strategy given a reward outcome on the previous trial, and P(R) is the probability of reward. In the equation above, H(Str) is the Shannon entropy of strategy, measuring the randomness in choice strategy in terms of stay or switch.(4)$$H\left(Str\right)=-\sum_{Str\in \{\mathrm{stay},\mathrm{switch}\}}P\left(Str\right)\log_{2}P\left(Str\right)=-\left(P\left(\mathrm{stay}\right)\cdot \log_{2}P\left(\mathrm{stay}\right)+P\left(\mathrm{switch}\right)\cdot \log_{2}P\left(\mathrm{switch}\right)\right),$$Here, P(Str) is the probability of staying or switching (P(stay)=1−P(switch)). $I(R_{t-1};Str_{t})$, which we refer to as MIRS, is the mutual information between reward outcome and strategy, equal to(5)$$\begin{array}{cc} \mathrm{MIRS} & =I(R_{t-1};Str_{t})\\ \; & =\sum_{R\in \{\mathrm{win},\mathrm{loss}\}}\sum_{Str\in \{\mathrm{stay},\mathrm{switch}\}}P\left(R,Str\right)\log_{2}\frac{P(R,Str)}{P(R)P(Str)}. \end{array}$$

Analogously, the conditional entropy of option-dependent strategy (EODS) represents the remaining uncertainty in the agent’s strategy after accounting for the choice made on the previous trial. It is defined as the difference between H(Str) and the mutual information between the previous choice and strategy (MIOS), as follows:(6)$$\begin{array}{cc} \mathrm{EODS} & =H(Str_{t}|Opt_{t-1})\\ \; & =H(Str)-I(Opt;Str)\\ \; & =-\sum_{Opt\in \{\mathrm{better},\mathrm{worse}\}}\sum_{Str\in \{\mathrm{stay},\mathrm{switch}\}}P\left(Opt,Str\right)\log_{2}\frac{P(Opt,Str)}{P(Opt)}, \end{array}$$
where $Opt_{t-1}$ indicates the option chosen on the previous trial, with 1 indicating the better option and 0 indicating the worse option, as defined by the assigned reward probabilities, and P(Opt) is the probability of choosing the better option. $I(Opt_{t-1};Str_{t})$ denotes the mutual information between the chosen option and the agent’s strategy, referred to as MIOS, and is calculated as follows:(7)$$\begin{array}{cc} \mathrm{MIOS} & =I(Opt_{t-1};Str_{t})\\ \; & =\sum_{Opt\in \{\mathrm{better},\mathrm{worse}\}}\sum_{Str\in \{\mathrm{stay},\mathrm{switch}\}}P\left(Opt,Str\right)\log_{2}\frac{P(Opt,Str)}{P(Opt)P(Str)}. \end{array}$$

To consider the combined effect of reward outcome and chosen option (i.e., winning or losing after choosing the better or worse option), we also considered a generalized metric that quantifies the combined effects of the two variables. Specifically, we define the conditional entropy of reward- and option-dependent strategy (ERODS) as follows:(8)$$\begin{array}{cc} \mathrm{ERODS} & =H(Str_{t}|R_{t-1},Opt_{t-1})\\ \; & =H(Str)-I(R,Opt;Str)\\ \; & =-\sum_{R\in \{\mathrm{win},\mathrm{loss}\}}\sum_{Opt\in \{\mathrm{better},\mathrm{worse}\}}\sum_{Str\in \{\mathrm{stay},\mathrm{switch}\}}P\left(R,Opt,Str\right)\log_{2}\frac{P(R,Opt,Str)}{P(R,Opt)}. \end{array}$$Here, I(R,Opt;Str) denotes the mutual information between the combination of the previous reward and choice outcomes (R,Opt) and subsequent choice strategy (MIROS):(9)$$\begin{array}{cc} \mathrm{MIROS} & =I(R_{t-1},Opt_{t-1};Str_{t})\\ \; & =\sum_{R\in \{\mathrm{win},\mathrm{loss}\}}\sum_{Opt\in \{\mathrm{better},\mathrm{worse}\}}\sum_{Str\in \{\mathrm{stay},\mathrm{switch}\}}P\left(R,Opt,Str\right)\log_{2}\frac{P(R,Opt,Str)}{P(R,Opt)P(Str)}. \end{array}$$

#### 2.2.1. Decomposition of Information-Theoretic Metrics

For the metrics described above, we also analyzed their decompositions into component values associated with each specific outcome of the conditioning variable. In the case of conditional entropy measures, this corresponds to the conditional entropy of *Y* given a specific value of X=x, weighted by the probability P(x). For instance, ERDS can be decomposed into two components based on previous reward outcomes, $ERDS_{+}$ and $ERDS_{-}$, which reflect the uncertainty reduction following rewarded (win) and unrewarded (loss) trials, respectively [32]:(10)$$\begin{array}{cc} {\mathrm{ERDS}}_{+} & =H\left(Str_{t}|R_{t-1}=\mathrm{win}\right)\cdot P\left(\mathrm{win}\right)\\ \; & =-\left(P\left(\mathrm{stay},\mathrm{win}\right)\cdot \log_{2}\frac{P(\mathrm{stay},\mathrm{win})}{P(\mathrm{win})}+P\left(\mathrm{switch},\mathrm{win}\right)\cdot \log_{2}\frac{P(\mathrm{switch},\mathrm{win})}{P(\mathrm{win})}\right), \end{array}$$(11)$$\begin{array}{cc} {\mathrm{ERDS}}_{-} & =H\left(Str_{t}|R_{t-1}=\mathrm{loss}\right)\cdot P\left(\mathrm{loss}\right)\\ \; & =-\left(P\left(\mathrm{stay},\mathrm{loss}\right)\cdot \log_{2}\frac{P(\mathrm{stay},\mathrm{loss})}{P(\mathrm{loss})}+P\left(\mathrm{switch},\mathrm{loss}\right)\cdot \log_{2}\frac{P(\mathrm{switch},\mathrm{loss})}{P(\mathrm{loss})}\right). \end{array}$$This decomposition guarantees that ${\mathrm{ERDS}}_{+}+{\mathrm{ERDS}}_{-}=\mathrm{ERDS}$ (Equation (Equation 3)).

Similarly, EODS can be decomposed into $EODS_{B}$ and $EODS_{W}$ based on the choice of the better or worse option on the previous trial:(12)$$\begin{array}{cc} {\mathrm{EODS}}_{B} & =H\left(Str_{t}|Opt_{t-1}=\mathrm{better}\right)\cdot P\left(\mathrm{better}\right)\\ \; & =-\left(P\left(\mathrm{stay},\mathrm{better}\right)\cdot \log_{2}\frac{P(\mathrm{stay},\mathrm{better})}{P(\mathrm{better})}+P\left(\mathrm{switch},\mathrm{better}\right)\cdot \log_{2}\frac{P(\mathrm{switch},\mathrm{better})}{P(\mathrm{better})}\right), \end{array}$$(13)$$\begin{array}{cc} {\mathrm{EODS}}_{W} & =H\left(Str_{t}|Opt_{t-1}=\mathrm{worse}\right)\cdot P\left(\mathrm{worse}\right)\\ \; & =-\left(P\left(\mathrm{stay},\mathrm{worse}\right)\cdot \log_{2}\frac{P(\mathrm{stay},\mathrm{worse})}{P(\mathrm{worse})}+P\left(\mathrm{switch},\mathrm{worse}\right)\cdot \log_{2}\frac{P(\mathrm{switch},\mathrm{worse})}{P(\mathrm{worse})}\right). \end{array}$$

Finally, ERODS is decomposed into four components based on the combination of the previous reward and the choice option:(14)$$\begin{array}{cc} {\mathrm{ERODS}}_{B+} & =H\left(Str_{t}|R_{t-1}=\mathrm{win},Opt_{t-1}=better\right)\cdot P\left(\mathrm{win},\mathrm{better}\right)\\ \; & =-\left(P\left(\mathrm{stay},\mathrm{win},\mathrm{better}\right)\cdot \log_{2}\frac{P(\mathrm{stay},\mathrm{win},\mathrm{better})}{P(\mathrm{win},\mathrm{better})}\right.\\ \; & +\left.P\left(\mathrm{switch},\mathrm{win},\mathrm{better}\right)\cdot \log_{2}\frac{P(\mathrm{switch},\mathrm{win},\mathrm{better})}{P(\mathrm{win},\mathrm{better})}\right), \end{array}$$(15)$$\begin{array}{cc} {\mathrm{ERODS}}_{B-} & =H\left(Str_{t}|R_{t-1}=\mathrm{loss},Opt_{t-1}=better\right)\cdot P\left(\mathrm{loss},\mathrm{better}\right)\\ \; & =-\left(P\left(\mathrm{stay},\mathrm{loss},\mathrm{better}\right)\cdot \log_{2}\frac{P(\mathrm{stay},\mathrm{loss},\mathrm{better})}{P(\mathrm{loss},\mathrm{better})}\right.\\ \; & +\left.P\left(\mathrm{switch},\mathrm{loss},\mathrm{better}\right)\cdot \log_{2}\frac{P(\mathrm{switch},\mathrm{loss},\mathrm{better})}{P(\mathrm{loss},\mathrm{better})}\right), \end{array}$$(16)$$\begin{array}{cc} {\mathrm{ERODS}}_{W+} & =H\left(Str_{t}|R_{t-1}=\mathrm{win},Opt_{t-1}=worse\right)\cdot P\left(\mathrm{win},\mathrm{worse}\right)\\ \; & =-\left(P\left(\mathrm{stay},\mathrm{win},\mathrm{worse}\right)\cdot \log_{2}\frac{P(\mathrm{stay},\mathrm{win},\mathrm{worse})}{P(\mathrm{win},\mathrm{worse})}\right.\\ \; & +\left.P\left(\mathrm{switch},\mathrm{win},\mathrm{worse}\right)\cdot \log_{2}\frac{P(\mathrm{switch},\mathrm{win},\mathrm{worse})}{P(\mathrm{win},\mathrm{worse})}\right), \end{array}$$(17)$$\begin{array}{cc} {\mathrm{ERODS}}_{W-} & =H\left(Str_{t}|R_{t-1}=\mathrm{loss},Opt_{t-1}=\mathrm{worse}\right)\cdot P\left(\mathrm{loss},\mathrm{worse}\right)\\ \; & =-\left(P\left(\mathrm{stay},\mathrm{loss},\mathrm{worse}\right)\cdot \log_{2}\frac{P(\mathrm{stay},\mathrm{loss},\mathrm{worse})}{P(\mathrm{loss},\mathrm{worse})}\right.\\ \; & +\left.P\left(\mathrm{switch},\mathrm{loss},\mathrm{worse}\right)\cdot \log_{2}\frac{P(\mathrm{switch},\mathrm{loss},\mathrm{worse})}{P(\mathrm{loss},\mathrm{worse})}\right), \end{array}$$
where the subscripts refer to winning after choosing the better option (B+), losing after the better option (B−), winning after the worse option (W+), and losing after the worse option (W−). Note that ERODS can be alternatively decomposed based on either reward outcome or choice option alone, by adding the relevant components. For example, this metric can be decomposed based on whether the previous trial resulted in reward or no reward, as follows:(18)$$\begin{array}{c} {\mathrm{ERODS}}_{+}={\mathrm{ERODS}}_{B+}+{\mathrm{ERODS}}_{W+},\\ {\mathrm{ERODS}}_{-}={\mathrm{ERODS}}_{B-}+{\mathrm{ERODS}}_{W-}. \end{array}$$ Alternatively, ERODS can be decomposed based on whether the better or worse option was selected in the previous trial:(19)$$\begin{array}{c} {\mathrm{ERODS}}_{B}={\mathrm{ERODS}}_{B+}+{\mathrm{ERODS}}_{B-},\\ {\mathrm{ERODS}}_{W}={\mathrm{ERODS}}_{W+}+{\mathrm{ERODS}}_{W-}. \end{array}$$

For decomposing mutual information metrics, we utilize the general formulation given by(20)$$\begin{array}{cc} I(X=x;Y) & =H(Y)-H(Y|X=x)\\ \; & =-\sum_{y\in Y}P\left(y\right)\cdot \log_{2}P\left(y\right)+\sum_{y\in Y}P\left(y|x\right)\cdot \log_{2}P\left(y|x\right), \end{array}$$This quantity represents the information that a specific value X=x provides about *Y*, and is known as the specific information [65]. The expected value of specific mutual information over all values of *X* is equal to the mutual information between *X* and *Y*, given by (Equation (Equation 2)):(21)$$\begin{array}{cc} I(X;Y) & =H(Y)-H(Y|X)\\ \; & =\sum_{x\in X}P\left(x\right)H\left(Y\right)-\sum_{x\in X}P\left(x\right)H\left(Y|X=x\right)\\ \; & =\sum_{x\in X}P\left(x\right)\left\{H\left(Y\right)-H\left(Y|X=x\right)\right\}\\ \; & =\sum_{x\in X}P\left(x\right)I\left(X=x;Y\right) \end{array}$$

Unlike the mutual information I(X;Y), which is always nonnegative, the specific mutual information for a specific event *x*, I(X=x;Y), can take negative values when H(Y)<H(Y|X=x) [66]. Conceptually, negative values indicate that the outcome *x* is misleading—or misinformative—about *Y* [67].

Using this definition, we compute two components of MIRS, ${\mathrm{MIRS}}_{+}$ and ${\mathrm{MIRS}}_{-}$, corresponding to previously rewarded and unrewarded outcome, respectively:(22)$$\begin{array}{cc} {\mathrm{MIRS}}_{+} & =I\left(R_{t-1}=\mathrm{win};Str_{t}\right)\cdot P\left(\mathrm{win}\right)\\ \; & =\left\{H\left(Str\right)-H\left(Str_{t}|R_{t-1}=\mathrm{win}\right)\right\}\cdot P\left(\mathrm{win}\right), \end{array}$$(23)$$\begin{array}{cc} {\mathrm{MIRS}}_{-} & =I\left(R_{t-1}=\mathrm{loss};Str\right)\cdot P\left(\mathrm{loss}\right)\\ \; & =\left\{H\left(Str\right)-H\left(Str_{t}|R_{t-1}=\mathrm{loss}\right)\right\}\cdot P\left(\mathrm{loss}\right). \end{array}$$These include the decomposition terms for conditional entropy (Equations (Equation 10) and (Equation 11), which satisfy the identity ($\mathrm{MIRS}={\mathrm{MIRS}}_{+}+{\mathrm{MIRS}}_{-}$). Similarly, the decompositions for mutual information between the option chosen on the previous trial and strategy, MIOS, are defined as(24)$$\begin{array}{cc} {\mathrm{MIOS}}_{B} & =I\left(Opt_{t-1}=\mathrm{better};Str_{t}\right)\cdot P\left(\mathrm{better}\right)\\ \; & =\left\{H\left(Str\right)-H\left(Str_{t}|Opt_{t-1}=\mathrm{better}\right)\right\}\cdot P\left(\mathrm{better}\right), \end{array}$$(25)$$\begin{array}{cc} {\mathrm{MIOS}}_{W} & =I\left(Opt_{t-1}=\mathrm{worse};Str_{t}\right)\cdot P\left(\mathrm{worse}\right)\\ \; & =\left\{H\left(Str\right)-H\left(Str_{t}|Opt_{t-1}=\mathrm{worse}\right)\right\}\cdot P\left(\mathrm{worse}\right). \end{array}$$These terms satisfy $\mathrm{MIOS}={\mathrm{MIOS}}_{B}+{\mathrm{MIOS}}_{W}$. Lastly, the decompositions for MIROS are given as(26)$$\begin{array}{cc} {\mathrm{MIROS}}_{B+} & =I\left(R_{t-1}=\mathrm{win},Opt_{t-1}=\mathrm{better};Str_{t}\right)\cdot P\left(\mathrm{win},\mathrm{better}\right)\\ \; & =\left\{H\left(Str\right)-H\left(Str_{t}|R_{t-1}=\mathrm{win},Opt_{t-1}=\mathrm{better}\right)\right\}\cdot P\left(\mathrm{win},\mathrm{better}\right), \end{array}$$(27)$$\begin{array}{cc} {\mathrm{MIROS}}_{B-} & =I\left(R_{t-1}=\mathrm{loss},Opt_{t-1}=\mathrm{better};Str_{t}\right)\cdot P\left(\mathrm{loss},\mathrm{better}\right)\\ \; & =\left\{H\left(Str\right)-H\left(Str_{t}|R_{t-1}=\mathrm{loss},Opt_{t-1}=\mathrm{better}\right)\right\}\cdot P\left(\mathrm{loss},\mathrm{better}\right), \end{array}$$(28)$$\begin{array}{cc} {\mathrm{MIROS}}_{W+} & =I\left(R_{t-1}=\mathrm{win},Opt_{t-1}=\mathrm{worse};Str_{t}\right)\cdot P\left(\mathrm{win},\mathrm{worse}\right)\\ \; & =\left\{H\left(Str\right)-H\left(Str_{t}|R_{t-1}=\mathrm{win},Opt_{t-1}=\mathrm{worse}\right)\right\}\cdot P\left(\mathrm{win},\mathrm{worse}\right), \end{array}$$(29)$$\begin{array}{cc} {\mathrm{MIROS}}_{W-} & =I\left(R_{t-1}=\mathrm{loss},Opt_{t-1}=\mathrm{worse};Str_{t}\right)\cdot P\left(\mathrm{loss},\mathrm{worse}\right)\\ \; & =\left\{H\left(Str\right)-H\left(Str_{t}|R_{t-1}=\mathrm{loss},Opt_{t-1}=\mathrm{worse}\right)\right\}\cdot P\left(\mathrm{loss},\mathrm{worse}\right). \end{array}$$These terms quantify the shared information between each instance of a reward–option combination and the agent’s strategy. Alternative decompositions can be computed similarly to ERODS, as in Equations (Equation 18) and (Equation 19). Notably, these decompositions reveal how specific combinations of trial outcomes influence subsequent strategy, providing additional information beyond the MIRS and MIOS metrics.

#### 2.2.2. Normalization of Mutual Information Metrics

The mutual information metrics defined thus far quantify the total amount of shared information between the random variables *X* and *Y*. However, one might be interested in the *fraction* of uncertainty in *Y* that is reduced by the knowledge of *X*, rather than the absolute amount of shared information [68,69,70]. In our setting, if the agent’s strategy is largely deterministic (i.e., H(Str) is low), then the mutual information will also be low due to the small value of H(Str) itself (e.g., see the gray circle specifying H(Str) in Figure 1b). Therefore, to account for the baseline uncertainty in the strategy, we also considered normalized mutual information metrics, obtained by dividing the mutual information by H(Str).

More specifically, we define normalized mutual information metrics for MIRS, MIOS, and MIROS as follows:(30)$$\begin{array}{cc} n\text{-}\mathrm{MIRS} & =I\left(R_{t-1};Str_{t}\right)/H\left(Str\right), \end{array}$$(31)$$\begin{array}{cc} n\text{-}\mathrm{MIOS} & =I\left(Opt_{t-1};Str_{t}\right)/H\left(Str\right), \end{array}$$(32)$$\begin{array}{cc} n\text{-}\mathrm{MIROS} & =I\left(R_{t-1},Opt_{t-1};Str_{t}\right)/H\left(Str\right). \end{array}$$These metrics quantify the proportion of uncertainty in H(Str) that is explained by each corresponding mutual information term. The normalized decompositions for each metric are similarly obtained by dividing each component by H(Str). For example, normalized mutual information between reward and strategy is computed as(33)$$\begin{array}{cc} n\text{-}\mathrm{MIRS} & ={n\text{-}\mathrm{MIRS}}_{+}+{n\text{-}\mathrm{MIRS}}_{-},\\ \; & =\frac{P\left(\mathrm{win}\right)\cdot I\left(R_{t-1}=\mathrm{win};Str_{t}\right)}{H(Str)}+\frac{P\left(\mathrm{loss}\right)\cdot I\left(R_{t-1}=\mathrm{loss};Str_{t}\right)}{H(Str)}, \end{array}$$
and analogously, normalized mutual information between choice option and strategy is defined as(34)$$\begin{array}{cc} n\text{-}\mathrm{MIOS} & ={n\text{-}\mathrm{MIOS}}_{B}+{n\text{-}\mathrm{MIOS}}_{W},\\ \; & =\frac{P\left(\mathrm{better}\right)\cdot I\left(Opt_{t-1}=\mathrm{better};Str_{t}\right)}{H(Str)}+\frac{P\left(\mathrm{worse}\right)\cdot I\left(Opt_{t-1}=\mathrm{worse};Str_{t}\right)}{H(Str)}. \end{array}$$Finally, the normalized mutual information between reward–choice combination and strategy can be computed as follows (dropping subscripts for trials for simplicity):(35)$$\begin{array}{cc} n\text{-}\mathrm{MIROS} & ={n\text{-}\mathrm{MIROS}}_{B+}+{n\text{-}\mathrm{MIROS}}_{B-}+{n\text{-}\mathrm{MIOS}}_{W+}+{n\text{-}\mathrm{MIOS}}_{W-},\\ \; & =\frac{P(\mathrm{better},\mathrm{win})\cdot I(R=\mathrm{win},Opt=\mathrm{better};Str)}{H(Str)}+\frac{P(\mathrm{better},\mathrm{loss})\cdot I(R=\mathrm{loss},Opt=\mathrm{better};Str)}{H(Str)}\\ \; & +\frac{P(\mathrm{worse},\mathrm{win})\cdot I(R=\mathrm{win},Opt=\mathrm{worse};Str)}{H(Str)}+\frac{P(\mathrm{worse},\mathrm{loss})\cdot I(R=\mathrm{loss},Opt=\mathrm{worse};Str)}{H(Str)}. \end{array}$$

### 2.3. Reinforcement Learning Models and Simulations

We used multiple reinforcement learning (RL) models to simulate two variants of the probabilistic reversal learning task in order to illustrate that information-theoretic metrics can detect (1) the differential learning rates for rewarded versus unrewarded trials, (2) changes in learning rates due to metaplasticity, (3) the influence of reward harvest rate, and (4) the use of alternative learning strategies in multi-dimensional reward environments.

For each task, we generated synthetic choice data using a range of model parameters and computed a set of relevant behavioral metrics from the resulting data. Below, we outline the general architecture of the RL models used, which are based on the standard Q-learning algorithm [25,28,29,30]. In this algorithm, the reward values of the two choice options, $Q_{A}$ and $Q_{B}$, are updated on a trial-by-trial basis using reward feedback, and the difference between these values is used to make a choice (Figure 2a). Specifically, the value of the chosen option on a given trial *t*, $Q^{c}$, is updated according to the following rule:(36)$$\begin{array}{c} Q^{c}\left(t+1\right)=Q^{c}\left(t\right)+\alpha \left(R\left(t\right)-Q^{c}\left(t\right)\right), \end{array}$$
where R(t) is the binary reward outcome (1 if rewarded, 0 if unrewarded), and α denotes the learning rate, which determines the magnitude of the value update by scaling the reward prediction error, $RPE=R\left(t\right)-Q^{c}\left(t\right)$. The value of the unchosen option remains unchanged. All value estimates are initialized at 0.5 for the first trial.

On each trial, the choice is determined stochastically based on the probability of selecting each option, $P_{j}$, computed using the softmax function, as follows:(37)$$\begin{array}{c} P_{j}\left(t\right)=\frac{e^{\beta Q_{j}\left(t\right)}}{e^{\beta Q_{j}\left(t\right)}+e^{\beta Q_{k}\left(t\right)}}=\frac{1}{1+e^{-\beta (Q_{j}\left(t\right)-Q_{k}\left(t\right))}}, \end{array}$$
where the indices *j* and *k* correspond to the two choice alternatives, and β is the inverse temperature parameter that controls the sensitivity to value differences (i.e., slope of the softmax function). We selected the softmax function (referred to as the Fermi function in the evolutionary game theory literature [71]) because it is widely adopted in empirical research and follows naturally from normative rationality principles [72,73]. In its simplest form, the RL model has only two free parameters: α and β. Below, we describe the variants of the RL models used in each set of simulations.

#### 2.3.1. Simulations of Positivity Bias

In this set of simulations, we examined the positivity bias—also known as the optimism bias—which refers to the tendency of an agent to learn more from feedback with positive valence than from feedback with negative valence [44,45,46,47]. This bias is often quantified by estimating separate learning rates for the outcomes that either are “better” or “worse” than expected, or equivalently, for positive and negative prediction errors, in the context of error-driven learning. In the case of binary reward outcomes, these correspond to the learning rates following rewarded and unrewarded trials, respectively. Therefore, the learning rule in Equation (Equation 36) can be generalized as(38)$$Q^{c}\left(t+1\right)=\left\{\begin{array}{cc} \; & Q^{c}\left(t\right)+\alpha_{+}\left(R\left(t\right)-Q^{c}\left(t\right)\right),\;\mathrm{if}\;R\left(t\right)=1\\ \; & Q^{c}\left(t\right)+\alpha_{-}\left(R\left(t\right)-Q^{c}\left(t\right)\right),\;\mathrm{if}\;R\left(t\right)=0, \end{array}\right.$$
where $\alpha_{+}$ and $\alpha_{-}$ denote the learning rates used to update the value of the chosen option on rewarded and unrewarded trials, respectively. Positivity bias is thus formally characterized by the condition where $\alpha_{+}>\alpha_{-}$.

We used the above RL model to simulate the choice behavior in a probabilistic reversal learning task. Each block consisted of 80 trials with an 80/20 reward schedule, and a reversal in reward contingencies occurred after trial 40. To examine how positivity bias can be detected using information-theoretic metrics, we simulated choice behavior using different combinations of ($\alpha_{+}$, $\alpha_{-}$) values. We set the inverse temperature at β=10, consistent with values observed in previous experimental studies [44,74]. To explore a plausible range of learning rates while avoiding extreme values, we varied each learning rate $\alpha_{i}\in \left[0.1,0.9\right]$ in increments of 0.05, resulting in a 17-by-17 grid of ($\alpha_{+}$, $\alpha_{-}$) combinations. For each point in this parameter space, choice behavior was simulated over 10,000 blocks.

To test whether the information-theoretic measures contain sufficient information to determine the presence of positivity bias ($\alpha_{+}>\alpha_{-}$), we utilized a linear decoder based on logistic regression (implemented using the *fitclinear* function in MATLAB, version R2023a). Decoders were trained on metrics computed from the choice behavior of training samples and tested on a separate held-out set, with the goal of predicting whether the underlying RL model used to generate the data had $\alpha_{+}>\alpha_{-}$.

More specifically, in each decoding experiment, we randomly sampled RL agents, each assigned a true ($\alpha_{+}$, $\alpha_{-}$) pair drawn independently from uniform distributions over $\alpha_{i}\in \left[0.1,0.9\right]$ (Figure 3e). Twenty RL agents were assigned to each group, labeled as either “optimistic” ($\alpha_{+}$ > $\alpha_{-}$) or “pessimistic” ($\alpha_{+}$ < $\alpha_{-}$). For each agent, every simulated block (out of total of 100) used a distinct ($\alpha_{+}$, $\alpha_{-}$) pair sampled from Gaussian distributions (σ=0.03) centered on that agent’s true ($\alpha_{+}$, $\alpha_{-}$) values. For each block, we obtained theoretical averages of the information-theoretic metrics by averaging across 10 repeated simulations. To evaluate decoding performance, we employed a leave-one-out procedure at the agent level: the decoder was trained on the information-theoretic metrics computed from the choice behavior of all agents except the one being tested. The training data were balanced to include an equal number of optimistic and pessimistic agents. Decoding accuracy was averaged across 100 independent decoding experiments, each using a unique set of RL agents.

Finally, to benchmark our decoder against behavioral features beyond the information-theoretic metrics, we trained a separate decoder on the coefficients of a logistic regression predicting choice from past choice and reward history. More specifically, we fit the following logistic regression model [33] to the same segment of choice data:(39)$$\log\left(\frac{P(C(t)=j)}{P(C(t)=k)}\right)=\beta_{0}+\sum_{i=1}^{5}\beta_{i}^{R}\left(R_{j}\left(t-i\right)-R_{k}\left(t-i\right)\right)+\sum_{i=1}^{5}\beta_{i}^{U}\left(U_{j}\left(t-i\right)-U_{k}\left(t-i\right)\right),$$
where C(t) indicates the choice made on trial *t*, *j* and *k* index the two choice options, *R* indicates a rewarded choice (1 if rewarded, 0 otherwise), and *U* denotes an unrewarded choice (1 if unrewarded, 0 otherwise). We used the resulting ten regression coefficients ($\beta^{R}$ and $\beta^{U}$) to decode whether a given agent is optimistic or pessimistic.

#### 2.3.2. Simulations of Reward-Dependent Metaplasticity

In this set of simulations, we examined whether adjustments in the learning rates—predicted by reward-dependent metaplasticity [48,49]—can be detected using information-theoretic metrics. Importantly, reward-dependent metaplasticity enables dynamic adjustments of synaptic plasticity over time and can account for stimulus- or action-specific learning rates, including asymmetries such as positivity bias. Using the same probabilistic reversal learning task with an 80/20 reward schedule as in the positivity bias simulations, we compared the choice behavior of the metaplastic model to that of the ‘plastic’ model, which is equivalent to the standard RL model described earlier. Below, we briefly describe the metaplastic model (see [48] for more details).

Importantly, the standard RL algorithm can be implemented through binary synapses that encode and update the value of each choice option through reward-dependent plasticity: transitioning from a “weak” to a “strong” state following reward (reward-dependent potentiation), and from a “strong” to a “weak” state following no reward (reward-dependent depression) [49,75,76] (Figure 4a). Importantly, the proportion of synapses in the “strong” state provides an estimate of the value of a given choice option ($Q_{j}$), as this quantity increases and decreases following reward and no reward, respectively [75]. We assume that only the synapses associated with the chosen option undergo state transitions according to the reward outcome (reward-dependent plasticity), while those associated with the unchosen option remain unchanged—consistent with the assumption of a standard Q-learning model.

The metaplastic model generalizes the above reward-dependent plasticity mechanism by introducing multiple meta-states for each level of synaptic efficacy, with synapses occupying deeper meta-states being more resistant—or stable—with respect to future changes (Figure 4b). More specifically, synapses in the metaplastic model undergo both plastic and metaplastic transitions. During plastic transitions, synaptic efficacy shifts between “weak” and “strong” states. During metaplastic transitions, synaptic efficacy remains unchanged, but the stability is modified as the synapse transitions to a deeper or shallower meta-state (Figure 4b).

Formally, higher stability of deeper meta-states is captured by decreasing transition probabilities $q_{i}$ for the *i*^th^ meta-state, governed by a power law, as follows:(40)$$q_{i}=q_{1}^{\frac{i(m-2)+1}{m-1}}\;\mathrm{for}\;2\leq i\leq m,$$
where *m* is the number of meta-states, $q_{1}$ is the (baseline) transition probability between the most unstable weak and most unstable strong meta-states, and $q_{i}$ is the transition probability from weak (or strong) meta-state i+1 to the most unstable strong (or weak) state in response to positive (or negative) reward feedback (reward vs. no reward), as indicated by the diagonal arrows in Figure 4b.

Additionally, metaplastic synapses can undergo transitions that do not alter their synaptic efficacy (vertical arrows in Figure 4b), with the probability of such transitions decreasing for deeper meta-states:(41)$$p_{i}={p_{1}}^{i}\;\mathrm{for}\;2\leq i\leq m-1,$$
where $p_{i}$ indicate the transition probability between the *i*^th^ and the (i+1)^th^ meta-states. After positive reward feedback (potentiation events), “weak” synapses transition toward less stable meta-states while “strong” synapses transition toward more stable, deeper meta-states. Conversely, after negative reward feedback (depression events), “weak” synapses undergo transition toward more stable, deeper meta-states while “strong” synapses transition toward less stable meta-states (vertical arrows in Figure 4b).

The value of each option in the metaplastic model is computed by summing over the fractions of synapses in the strong states:(42)$$Q_{j}\left(t\right)=\sum_{i=1}^{m}S_{j,i}\left(t\right),$$
where *j* indexes the choice option and $S_{j,i}\left(t\right)$ indicates the fraction of synapses in the *i*^th^ strong meta-state on a given trial.

Because the metaplastic model includes multiple meta-states with different transition rates, its update in response to reward feedback—and thus its rate of learning—changes over time. This dynamic can be quantified using two “effective” learning rates computed on each trial. The effective learning rate following reward ($\alpha_{+}^{eff}$) is defined as the fraction of synapses that transition from weak to strong states, while the effective learning rate following no reward ($\alpha_{-}^{eff}$) reflects the fraction transitioning from strong to weak states, as follows:(43)$$\begin{array}{cc} \alpha_{+}^{eff}\left(t\right) & =\frac{Q^{c}\left(t+1\right)-Q^{c}\left(t\right)}{R\left(t\right)-Q^{c}\left(t\right)}\;\mathrm{if}\;R\left(t\right)=1,\\ \alpha_{-}^{eff}\left(t\right) & =\frac{Q^{c}\left(t+1\right)-Q^{c}\left(t\right)}{R\left(t\right)-Q^{c}\left(t\right)}\;\mathrm{if}\;R\left(t\right)=0, \end{array}$$
where the numerator indicates the change in the overall value of the chosen option, while the denominator corresponds to the prediction error.

To simulate choice behavior, we set α=0.3 and β=10 for the plastic model with a single learning rate. For the metaplastic model, we used the model with m=4 meta-states and set $p_{1}=0.4$ for the baseline meta-transition probability. We set the transition probability $q_{1}=0.516$, which yields an initial effective learning rate of 0.3 on the first trial of the block (Figure 4c). We also tested the choice behavior of a plastic model with differential learning rates for rewarded and unrewarded outcomes ($\alpha_{+}\neq \alpha_{-}$). These learning rates were estimated by fitting the plastic model to the choice behavior of the metaplastic model on each block via maximum-likelihood estimation (optimized with the *fmincon* function in MATLAB, version 2023a).

Similarly to the positivity bias simulations, we used a linear decoder to test whether the emergence of positivity bias ($\alpha_{+}>\alpha_{-}$)—driven by metaplasticity over time—can be detected using the information-theoretic metrics. The training procedure involved randomly drawing a set of ($\alpha_{+},\alpha_{-}$) values from the grid of [0.2, 0.4] in a step size of 0.01, under the assumption that the approximate range of α value is known (α=0.3 in our case). Each parameter space was labeled as either “optimistic” ($\alpha_{+}>\alpha_{-}$) or “pessimistic” ($\alpha_{+}<\alpha_{-}$). The points with $\alpha_{+}=\alpha_{-}$ were dropped from the training set. We then computed the decoder’s posterior probability on held-out samples, which were generated from the choice behavior of the three models: (1) a plastic model with a single learning rate ($\alpha_{+}=\alpha_{-}=\alpha$), (2) the metaplastic model, and (3) a plastic model with separate learning rates ($\alpha_{+}\neq \alpha_{-}$), estimated from the choice behavior of the metaplastic model.

#### 2.3.3. Simulations of Reward Harvest Rate Effects on Behavior

In this set of simulations, we examined whether the influence of the overall reward rate on a long time scale—referred to as reward harvest rate—can be detected using the information-theoretic metrics. We used the same probabilistic reversal learning task as in the positivity bias simulations and tested two RL models for generating choice behavior: (1) the standard RL model with a single learning rate, as described earlier, and (2) an augmented RL model that included an additional adjustment mechanism based on reward harvest rate (RL with RHR model; Figure 5a). The latter model served as the ground truth for a potential pathway through which reward harvest rate could influence choice strategy—specifically, by increasing the tendency to win–stay and lose–switch as the harvest reward rate increases. More specifically, the augmented model tracks a third variable *H*, representing the overall reward harvest rate independent of any specific choice option, and updates it as follows:(44)$$H\left(t+1\right)=H\left(t\right)+\alpha_{H}\left(R\left(t\right)-H\left(t\right)\right),$$
where $\alpha_{H}$ is the update rate for *H*. Therefore, *H* provides an exponentially weighted moving average of harvested rewards. For simplicity, we set $\alpha_{H}=\alpha =0.3$ in the simulations. In practice, although $\alpha_{H}$ can be directly optimized to the data, one can equally justify choosing an arbitrary $\alpha_{H}$ for behavioral analysis to avoid modeling assumptions and obtain model-independent measures. To incorporate a win–stay/lose–switch bias that is modulated by the reward harvest rate *H*, the choice probability in this model is computed as follows:(45)$$\begin{array}{c} P_{j}\left(t\right)=\frac{1}{1+e^{-\left[\beta \left(Q_{j}\left(t\right)-Q_{k}\left(t\right)\right)+\beta_{H}Bias_{j}\left(t\right)\right]}}, \end{array}$$
where $\beta_{H}$ is a free parameter controlling the overall influence of the win–stay/lose–switch bias term for option *j*, denoted as Bias. This bias term is calculated as follows:(46)$$\begin{array}{c} Bias_{j}\left(t\right)=C_{j}\left(t-1\right)\cdot \left(R(t-1)-H(t-1)\right), \end{array}$$
where $C_{j}$ indicates whether option *j* was chosen in the previous trial (1 if chosen, −1 otherwise), and the term R−H dynamically adjusts the agent’s tendency to repeat (after reward) or switch (after no reward) based on how much the recent reward outcome differs from reward harvest rate *H*. In our simulations, we set $\beta =10,\beta_{H}=2$.

To quantify the effect of reward harvest rate on choice strategy, we introduced an additional mutual information metric, referred to as MIRHS, which captures the joint influence of previous reward outcome and the “reward harvest state” (as defined by reward rate) on the agent’s strategy, as follows:(47)$$\begin{array}{cc} \mathrm{MIRHS} & =I(R_{t-1},RHS_{t-1};Str_{t})\\ \; & =\sum_{R\in \{\mathrm{win},\mathrm{loss}\}}\sum_{RHS\in \{\mathrm{high},\mathrm{low}\}}\sum_{Str\in \{\mathrm{stay},\mathrm{switch}\}}P\left(R,RHS,Str\right)\log_{2}\frac{P(RHS,Str)}{P(R,RHS)P(Str)}, \end{array}$$
where RHS denotes the reward harvest state—a binary, discretized variable indicating whether the reward harvest rate *H* on a given (previous) trial was classified as “high” or “low”, based on the median split within the current block (Figure 5b) [37]. We also used a normalized version of this metric, n-MIRHS $=I\left(R_{t-1},RHS_{t-1};Str_{t}\right)/H\left(Str_{t}\right)$, which is decomposed as follows (dropping subscripts for trials for simplicity):(48)$$\begin{array}{cc} n\text{-}\mathrm{MIRHS} & ={n\text{-}\mathrm{MIRHS}}_{high+}+{n\text{-}\mathrm{MIRHS}}_{high-}+{n\text{-}\mathrm{MIRHS}}_{low+}+{n\text{-}\mathrm{MIRHS}}_{low-},\\ \; & =\frac{P(\mathrm{high},\mathrm{win})\cdot I(R=\mathrm{win},RHS=\mathrm{high};Str)}{H(Str)}+\frac{P(\mathrm{high},\mathrm{loss})\cdot I(R=\mathrm{loss},RHS=\mathrm{high};Str)}{H(Str)}\\ \; & +\frac{P(\mathrm{low},\mathrm{win})\cdot I(R=\mathrm{win},RHS=\mathrm{low};Str)}{H(Str)}+\frac{P(\mathrm{low},\mathrm{loss})\cdot I(R=\mathrm{loss},RHS=\mathrm{low};Str)}{H(Str)}, \end{array}$$
where the subscripts of the metric refer to winning during a high-RHS state (high+), losing during a high-RHS state (high−), winning during a low-RHS state (low+), and losing during a low-RHS state (low−). These metrics therefore separate the influence of immediate, reward outcome from that of the reward harvest state. To isolate the overall effect of RHS, we also computed an alternative decomposition, as follows:(49)$$\begin{array}{c} {\mathrm{MIRHS}}_{high}={\mathrm{MIRHS}}_{high+}+{\mathrm{MIRHS}}_{high-}, \end{array}$$(50)$$\begin{array}{c} {\mathrm{MIRHS}}_{low}={\mathrm{MIRHS}}_{low+}+{\mathrm{MIRHS}}_{low-}, \end{array}$$

#### 2.3.4. Simulations of Learning in Multidimensional Reward Environments

To investigate the presence of alternative learning strategies in the agent’s behavior, we conducted a final set of simulations using a variant of the probabilistic reversal learning task in which different attributes of the choice options predicted reward outcomes at different time points.

In this task, at any given point within a block of trials, only one of the two choice attributes—the shape and the color of the stimuli—is predictive of reward probabilities (80% vs. 20%). Initially, the reward schedule for a given block is assigned to the color of the stimuli regardless of their shape, with higher reward probability assigned to either green or orange objects. Therefore, unlike the reversal learning task used in the previous simulations, this task involves two types of reversals: (i) reversal in the reward probability, as previously considered, and (ii) reversal in the feature predictive of reward (Figure 6a). Therefore, in a *Color-to-Color* (or similarly *Shape-to-Shape*) block, the reward contingency was reversed between the better and worse values of the same attribute—for example, between two colors (Figure 6c). In contrast, in a *Color-to-Shape* block, the predictive attribute switched, and reward became associated with the shape of the object (e.g., triangle = 80% and square = 20%), regardless of its color (Figure 6d). The identity of the block type was unknown to the agent, and the two block types were randomly interleaved throughout the session. The agent therefore had to adapt their learning and choice strategies solely based on the reward feedback they received.

To capture the strategies that more realistic agents might adopt, we used a generalized RL model that dynamically arbitrates between competing learning strategies. To that end, we used a variant of the model introduced by Woo et al. [40], which simultaneously tracks the values of multiple choice attributes (Figure 6b). In this model, the value estimates of chosen and unchosen options for each attribute are updated simultaneously as follows:(51)$$\begin{array}{cc} Q_{i}^{c}\left(t+1\right) & =Q_{i}^{c}\left(t\right)+\alpha \left(R\left(t\right)-Q_{i}^{c}\left(t\right)\right), \end{array}$$(52)$$\begin{array}{cc} Q_{i}^{u}\left(t+1\right) & =\left(1-\gamma \right)Q_{i}^{u}\left(t\right), \end{array}$$
where *i* indexes the choice attribute (Color or Shape), and *c* indexes the chosen option within each attribute (i.e., Color ∈{green,orange} and Shape ∈{triangle,square}). The index *u* refers to the unchosen option within each attribute, and γ denotes the decay rate of value for unchosen options. Note that this decay mechanism is biologically motivated and included to account for the increased number of choice features in this model, reflecting the brain’s limited capacity to retain the memory of all value estimates. To generate a choice, the model first computes the overall value, *V*, of the two choice alternatives through a linear weighted combination of the attributes, as follows:(53)$$\begin{array}{c} V_{j}\left(t\right)=Q_{Color,j}\times \omega \left(t\right)+Q_{Shape,j}\times \left(1-\omega \left(t\right)\right), \end{array}$$
where ω(t) is the arbitration weight on trial *t* specifying the relative contribution of the color attribute to overall value, and the index *j* denotes the choice options to the left and right of the screen (for example, if the leftward option is a green triangle, $V_{left}\left(t\right)=Q_{green}\left(t\right)\omega \left(t\right)+Q_{triangle}\left(t\right)\left(1-\omega \left(t\right)\right)$). The difference in overall values *V* is then used to compute choice probability, similarly to Equation (Equation 37). It is important to note that a linear combination of feature values to an overall value is used primarily for simplicity in describing the model and does not necessarily imply that such an integrated value is explicitly constructed. In practice, it is more plausible that the value of each attribute is compared directly—albeit with different weights—when making choices [77].

To dynamically control the arbitration weight between competing strategies, we considered the following update rule:(54)$$\omega \left(t+1\right)=\left\{\begin{array}{cc} \; & \omega \left(t\right)+\alpha_{\omega }\left|\Delta Rel\left(t\right)\right|\left(1-\omega \left(t\right)\right),\;\mathrm{if}\;\Delta Rel\left(t\right)>0\\ \; & \omega \left(t\right)+\alpha_{\omega }\left|\Delta Rel\left(t\right)\right|\left(0-\omega \left(t\right)\right),\;\mathrm{if}\;\Delta Rel\left(t\right)<0, \end{array}\right.$$
where ΔRel specifies the difference in the reliability between color and shape attributes in predicting rewards. Intuitively, the model assigns higher weight to the attribute which is estimated to have higher reliability for the given trial. We defined reliability based on the absolute value of the reward prediction error (RPE) for each attribute. Specifically, the reliability of an attribute is defined as inversely proportional to the magnitude of its RPE—a lower RPE (i.e., less surprise) indicates higher reliability [38]. Based on this definition, the reliability difference ΔRel is given by(55)$$\begin{array}{cc} \Delta Rel(t) & =|RPE_{Shape}|-|RPE_{Color}|,\\ \; & =|R\left(t\right)-Q_{Shape}^{c}|-|R\left(t\right)-Q_{Color}^{c}|, \end{array}$$
where $RPE_{Color}$ and $RPE_{Shape}$ refer to the reward prediction error between actual and predicted reward outcomes based on color or shape attribute, respectively. For example, if the color attribute yields a smaller RPE magnitude than the shape attribute, then ΔRel>0, and the model increases ω to bias decision making toward the color attribute on the next trial.

We considered three distinct decision-making strategies, each implemented as a special case of the arbitration model described above. In the first case, the model fixed ω=1 for all trials, representing an agent who relies exclusively on the color attribute. In the second case, ω=0.5 was fixed, modeling an agent who assigns equal weight to both color and shape attributes. In the third case, the full arbitration mechanism was implemented, with the agent dynamically adjusting ω based on the relative reliability of the two attributes in predicting reward. For this model, ω was initialized at 0.5 on the first trial. For the full model, we used the following parameter values: α=0.4, β=10, γ=0.2, $\alpha_{\omega }=0.4$.

To quantify the relative dominance of color-based versus shape-based learning strategies, we computed the conditional entropy of reward-dependent strategy (ERDS) separately for each attribute. Specifically, we defined stay/switch strategies separately for color and shape—denoted as $Str_{Color}$ and $Str_{Shape}$, respectively—based on whether the agent repeats or switches their choice option with respect to each attribute after reward feedback. The corresponding ERDS measure for each attribute was then defined as follows:(56)$$\begin{array}{cc} {\mathrm{ERDS}}_{Color} & =H(Str_{Color}|R),\\ {\mathrm{ERDS}}_{Shape} & =H(Str_{Shape}|R), \end{array}$$

To quantify the relative dominance of strategies in response to reward feedback, we computed the difference $\Delta ERDS=ERDS_{Shape}-ERDS_{Color}$. A higher value of ΔERDS indicates greater reliance on a color-based choice strategy.

## 3. Results

In the following sections, we illustrate the utility of information-theoretic metrics for identifying distinct learning and decision-making mechanisms using simulated data described above. We begin by presenting example behavior of an RL agent with a single learning rate, shown for three values: α={0.2,0.4,0.6} (Figure 2). The time course of the performance—defined as the probability of choosing the more rewarding option P(Better)—shows that, for all three learning rates, performance peaks within approximately 10 to 20 trials but more quickly for larger learning rates. However, the overall performance, calculated across the entire block, does not significantly differ among the three learning rates (rank-sum test, p>0.05; inset in Figure 2b).

Compared to the performance, the information-theoretic metrics exhibited more distinct trends across the different values of α (Figure 2c–f). For example, the entropy of strategy, H(Str), was highest for the RL agent simulated with α=0.2, followed by α=0.6 and α=0.4, which did not differ significantly from each other (inset in Figure 2c). Moreover, the trajectory of H(Str), which measures the entropy of stay/switch strategy, decreased over trials within a block and peaked following the reversal—when the better and worse options were swapped. The normalized mutual information between reward and strategy (n-MIRS) revealed that information gained from reward feedback was highest for α=0.6, followed by 0.4, and lowest for 0.2 (Figure 2d), consistent with the interpretation of the learning rate as a measure of how strongly choices are adjusted based on reward feedback. The normalized mutual information between choice option and strategy (n-MIOS) indicated that the agents gradually learned to identify and persist with the better option over trials, as reflected by the increasing P(Better) and decreasing H(Str) (Figure 2e). Finally, the normalized mutual information between option–reward combinations and strategy (n-MIROS) revealed that, despite differences in their temporal trajectories, the overall information shared between these variables plateaued at similar levels across all three learning rates (Figure 2f). However, the distinct temporal dynamics led to significant differences among agents as reflected in the average metric values computed over the entire block (inset in Figure 2f).

Overall, this example illustrates that information-theoretic metrics can detect subtle variations in underlying mechanisms—such as variations in the learning rates—that simpler measures like performance may fail to capture (compare the inset in Figure 2b with those in Figure 2c–f). In the following four sets of simulations, we further show how these information-theoretic metrics can differentiate between the choice behavior of various RL models used as ground truths.

### 3.1. Revealing Positivity Bias

In the first set of simulations, we examined whether positivity bias—formally defined as a larger learning rate for positive outcomes compared to negative outcome ($\alpha_{+}>\alpha_{-}$)—can be captured using information-theoretic metrics. To that end, we used a standard RL (Figure 3a) to simulate choice behavior in a probabilistic reversal learning task under different reward schedules: 80/20 and 40/10. We explored different combinations of $\alpha_{+}$ and $\alpha_{-}$ values to examine how asymmetric learning rates influence behavior across these environments.

We found that higher positivity bias (larger $\Delta \alpha =\alpha_{+}-\alpha_{-}$) resulted in higher performance across the two reward environments, especially for the 40/10 schedule (Pearson’s correlation between Δα and P(Better); 40/10: *r* = 0.904, *p* = 4.32 × 10^−108^; 80/20: *r* = 0.702, *p* = 2.88 × 10^−44^) (Figure 3b,c). This is consistent with the previous literature [30,46,47,78], demonstrating that positivity bias can improve performance in terms of reward harvest.

Because learning rates control how value estimates are updated in response to reward feedback, we hypothesized that information-theoretic metrics related to reward—specifically ERDS and n-MIRS—would be sufficient to detect positivity bias in the choice behavior. Following this intuition, we first compared the choice behavior of the standard RL agents simulated with selected values of $\alpha_{+}$ and $\alpha_{-}$ that exemplify “optimistic” ($\alpha_{+}>\alpha_{-}$), “neutral” ($\alpha_{+}=\alpha_{-}$), and “pessimistic” ($\alpha_{+}<\alpha_{-}$) tendencies (Figure 3c).

We found that the entropy of stay/switch strategy conditioned on reward (ERDS) was highest for the most pessimistic agent (cyan lines in Figure 3d), whereas the normalized mutual information between reward and strategy (n-MIRS) was lowest for the same agent (cyan lines in Figure 3e). However, metrics computed from the neutral agent (black lines) and the two optimistic agents (red and magenta lines) were less distinguished and plateaued at similar values. When simulating the full grid of $\alpha_{+}$ and $\alpha_{-}$ values, we observed that ERDS was significantly correlated with Δα, with higher positivity bias associated with lower ERDS (*r* = −0.778, *p* = 6.25 × 10^−60^; Figure 3f). In contrast, n-MIRS was only weakly correlated with Δα (*r* = −0.137, *p* = 0.020), and the direction of this relationship was opposite to that observed in the example shown in Figure 3e, suggesting an overall nonlinear relationship. Therefore, while a stronger positivity bias is generally associated with lower ERDS, the absolute values of these metrics alone do not reliably indicate whether a given behavior reflects positivity bias.

To identify candidate metrics that may be more informative of positivity bias, we next examined the decomposition of the above metrics based on reward outcome (reward vs. no reward). Intuitively, there is greater uncertainty about whether the agents will stay or switch following unrewarded trials compared to rewarded ones. Over time, agents learn to maximize reward by adopting a stay-dominated strategy following rewarded trials, as they continue selecting the better option. In contrast, after unrewarded trials, agents are more likely to switch. However, if the chosen option still has a relatively high reward probability (i.e., it is the better option), the likelihood of staying after a no-reward trial can also increase over time.

Consistent with this notion, we found that the trajectory of the decomposition of mutual information metric following no reward (n-MIRS- in Figure 3g) became negative over time, suggesting that negative reward outcomes became less informative regarding the agents’ subsequent strategy (i.e., H(Str)<H(Str|R=loss)). However, during the early portion of the block, unrewarded trials were initially informative about choice strategy, as indicated by the positive n-MIRS- values. Optimistic and pessimistic agents exhibited distinct temporal dynamics in how quickly this metric shifted toward negative values (shaded gray area in Figure 3g). When computed over the first 10 trials, the sign of the n-MIRS– metric was approximately predictive of whether an agent exhibited positivity bias (Figure 3g inset). Specifically, more optimistic agents exhibited more negative n-MIRS- values, indicating that their tendency to update more strongly after rewarded than unrewarded trials. This makes the no-reward trials less informative of their subsequent strategy. This pattern is consistent with the results shown in Figure 3b, where optimistic agents achieved better performance. To confirm this effect across the full parameter space, we computed n-MIRS- as a function of Δα and found a strong linear relationship in the 80/20 environment (*r* = −0.926, *p* = 1.27 × 10^−123^), as well as in the other two tested reward environments (60/40 and 40/10; Figure 3h). These results suggest that the decomposed mutual information measures, specifically n-MIRS, computed from the early period of each block, can be used to predict the presence of positivity bias. To test this idea, we conducted a classification analysis in which we randomly sampled groups of optimistic and pessimistic agents and used the information-theoretic metrics to predict whether a given metric profile originated from an optimistic agent using a linear decoder (Figure 3i; see Section 2.3.1 for details).

We found that the n-MIRS metric alone, when computed from the first 10 trials, yielded high cross-validated decoding accuracy across different environments (Figure 3j), significantly exceeding chance level (signed-rank test against 0.50; p<0.001 for all environments). Including all available metrics in the decoder led to a small but significant improvement in the decoding accuracy (signed-rank test; p=0.0011 for 50/10, p<0.001 for all other environments). In comparison, a decoder based on logistic regression weights achieved comparable accuracies (white bars in Figure 3j), indicating that the information-theoretic metrics contain comparable information to traditional regression-based measures—while offering more interpretable connections to underlying learning and decision-making processes. Overall, these results illustrate that information-theoretic metrics contain sufficient information to determine the presence of positivity bias when the ground truths are provided.

### 3.2. Revealing Reward-Dependent Metaplasticity

One potential neural mechanism underlying differential learning rates is reward-dependent metaplasticity, which generates dynamic learning rates by naturally adapting to the recent history of rewards in the environment [48,49,75,79]. To test whether such dynamic changes in the learning rates can be detected using information-theoretic metrics, we implemented a variant of the metaplastic model proposed by Farashahi et al. [48]. In our simulations, we assumed that each of the two choice options is assigned a set of metaplastic synapses which undergo transitions in response to reward feedback (Figure 4b; Section 2.3.2). By computing the “effective” learning rate—defined as the overall rate of value update on each trial—we found that the metaplastic model exhibited diverging learning rates following rewarded and unrewarded trials, with $\alpha_{+}>\alpha_{-}$ (solid lines in Figure 4c), consistent with previous findings [48]. The difference between the effective $\alpha_{+}$ and $\alpha_{-}$ increased over time, and plateaued just before the reversal (red solid curve in Figure 4c), driven primarily by an increase in $\alpha_{+}$ rather than a decrease in $\alpha_{-}$. This happens because, as more reward is obtained, the “weak” synapses encoding the better option mostly occupy the most unstable meta-states $w_{1}$ (Figure 4b), which have the highest transition probability $\alpha_{1}$ toward the “strong” state $s_{1}$, resulting in a high effective $\alpha_{+}$.

Using the choice behavior of this metaplastic agent as ground truth, we fitted a ’plastic’ model (standard RL) with differential learning rates to the simulated choice data from each block to test whether the fitted parameters reflected the pattern $\alpha_{+}>\alpha_{-}$ (noting that these learning rates are constant by definition in this model). The results confirmed this pattern (red and blue dashed lines in Figure 4c), suggesting that empirically observed trends of $\alpha_{+}>\alpha_{-}$ in previous studies may, in fact, reflect underlying reward-dependent metaplasticity.

To test whether the distinct learning mechanisms of the plastic and metaplastic models can be identified from choice behavior, we simulated the choice behavior of a metaplastic model and two plastic models, and then computed information-theoretic metrics from the simulated behavior. Specifically, we compared metrics across three models: (1) a plastic model with a single α=0.3 (plastic 1-α), (2) a metaplastic model initialized with the same effective learning rate of 0.3 on the first trial, and (3) a plastic model with differential learning rates (plastic 2-α) estimated from the metaplastic agent.

Comparison of the entropy of choice strategy indicated that the metaplastic agent was more consistent in its strategy compared to both plastic agents (rank-sum test on H(Str): vs. 1-α: *p* = 4.43 × 10^−28^; vs. 2-α: *p* = 2.43 × 10^−24^; Figure 4d). Moreover, the n-MIOS metric revealed that the metaplastic agent had the advantage of increasing the mutual information between the choice of the better option and subsequent strategy (Figure 4e), reflected in its superior performance relative to both plastic models (rank-sum test on n-MIOS: vs. 1-α: *p* = 5.58 × 10^−4^; vs. 2-α: *p* = 0.00280). The two plastic agents (1-α and 2-α) were less distinguishable from each other in terms of H(Str) (*p* = 0.0120), n-MIOS (*p* = 0.683), and overall performance (*p* = 0.848).

These results suggest that, although the parameters of the plastic 2-α model were directly estimated from the metaplastic agent, differences in the underlying learning mechanisms still give rise to distinct patterns in the information-theoretic metrics. This highlights the potential utility of such metrics in model recovery and validation; for example, by assessing whether a given candidate RL model can reproduce the information-theoretic metrics observed in the empirical data [32].

Given the observed differences between plastic and metaplastic models, we next quantified the extent to which the information-theoretic metrics discriminate between different models. We hypothesized that these metrics could be used to decode changes in the learning rates corresponding to positivity bias, using an approach similar to that employed in the previous section. To obtain the highest accuracy, we utilized all available information-theoretic metrics and their decompositions to train a linear decoder to discriminate between optimistic and pessimistic agents (see Section 2.3.2 in *Materials and Methods* for more details). We then applied this decoder to the metrics generated by the three models described above and computed the posterior probability that the decoder would classify the behavior as exhibiting positivity bias.

We found that the posterior probabilities derived from the metaplastic model (magenta line in Figure 4f) exhibited temporal dynamics closely resembling those of effective learning rates (Figure 4c). Specifically, as trials progressed, the decoder increasingly indicated that the behavior originated from optimistic agents with $\alpha_{+}>\alpha_{-}$. In contrast, the posterior probabilities for the two plastic agents remained relatively constant across the block. The plastic 1-α agent (black line in Figure 4f) maintained posterior probabilities at chance level, as it was neither optimistic nor pessimistic by design. On the other hand, the choice behavior of the plastic 2-α agent (gray line in Figure 4f) was classified as being optimistic, consistent with its estimated learning rates reflecting $\alpha_{+}>\alpha_{-}$ (dashed lines in Figure 4c).

### 3.3. Revealing Behavioral Adjustments Due to Reward Harvest Rate

Next, we investigated how information-theoretic metrics can be used to detect the influence cumulative overall reward feedback on all options (i.e., reward harvest rate) on an agent’s choice strategy. To that end, we used two types of RL model to generate choice behavior: (1) the standard RL with a single learning rate α and (2) the augmented RL model incorporating a reward harvest rate component (Figure 5a). Briefly, the augmented model tracks an average reward harvest rate *H*, which was not tied to any specific choice option, and uses it to modulate the agent’s win–stay and lose–switch tendencies (see Section 2.3.3 in *Materials and Methods* for more details). To quantify the effect of reward harvest rate on choice behavior, we defined a discrete variable referred to as the *reward harvest state* (RHS)—a binary indicator of whether the reward rate on a given trial was above or below the block median [37].

As shown for an example block, a high or low reward harvest state (RHS) did not consistently coincide with reward (or no reward) on the previous trial (Figure 5b), given the weak correlation between the reward harvest rate and binary reward feedback (*r* = 0.107, *p* = 0.343). Exploiting this property, we used RHS as an additional conditioning variable for the strategy Str, and measured the joint influence of previous reward and RHS on the agent’s subsequent strategy using the mutual information metric n-MIRHS (see Section 2.3.3 for more details). The trajectory of this metric revealed that overall information was significantly higher for the agent with the additional reward harvest rate component (*RL with RHR* agent) compared to the standard RL agent (rank-sum test, *p* = 2.56 × 10^−34^), reflecting the added influence of RHS on choice strategy Figure 5c.

To gain insight on how the high- and low-RHS states influence behavior, we next examined the decomposition of n-MIRHS based on RHS, corresponding to the influence of high (n-${\mathrm{MIRHS}}_{high}$, Figure 5d) and low RHS (n-${\mathrm{MIRHS}}_{low}$, Figure 5e). We found that the shared information between the high state and choice strategy was overall comparable across models but significantly higher for the standard RL agent than for the *RL with RHR* agent (difference in mean n-${\mathrm{MIRHS}}_{high}$ = 0.0445; rank-sum test, *p* = 4.67 × 10^−34^). In contrast, n-${\mathrm{MIRHS}}_{low}$ revealed a more pronounced distinction; it was significantly positive for the *RL with RHR* agent (signed-rank test, *p* = 3.90 × 10^−18^), but not significantly different from zero for the standard RL model (signed-rank test, *p* = 0.064; mean = 0.0019; inset in Figure 5e).

These results indicate that during the *high* state—when the reward rate was higher than usual due to successful learning—both models showed positive mutual information between the high RHS and subsequent strategy. In contrast, during the *low* state—when the agents were receiving rewards at a lower-than-usual rate—RHS was not informative about whether the standard RL agent would stay or switch on the next trial. This was not the case for the *RL with RHR* agent, for which n-${\mathrm{MIRHS}}_{low}>0$. Further decompositions based on reward feedback revealed that this effect was driven by a larger magnitude of n-${\mathrm{MIRHS}}_{low+}$ (following rewarded trials) compared to n-${\mathrm{MIRHS}}_{low-}$ (following unrewarded trials; compare red and blue bars in the Figure 5e inset for the RL with RHR agent). This pattern reflects an overall positive shift in information content resulting from the architecture of the RL with RHR model, which modulates stay/switch tendency as a function of the reward harvest rate *H*.

Lastly, to gain further insight into how the n-MIRHS metric rely on the model parameters, we simulated the choice behavior of the model with additional modulation by reward harvest rate using the full range of α and $\beta_{H}$ values. Using standardized regression, we then examined the contribution of the $\beta_{H}$ parameter, which determines the influence of reward harvest rate on choice behavior ($\beta_{H}=0$ corresponds to the standard RL). We found that, consistent with the example in (Figure 5c), larger values of $\beta_{H}$ predicted overall higher values of n-MIRHS (standardized regression of n-MIRHS on $\beta_{H}$, α, and their interaction: $b_{1}$ = 0.366, *p* = 3.13 × 10^−216^). Similar regression analyses on each of the decompositions yielded results consistent with Figure 5d,e. More specifically, $\beta_{H}$ did not significantly affect the n-MIRHS decomposition for the high-reward state (standardized coefficient predicting n-${\mathrm{MIRHS}}_{high}$: $b_{1}$ = 0.0150, *p* = 0.329) nor its further reward-based decompositions (n-${\mathrm{MIRHS}}_{high+}$: $b_{1}$ = 0.024, *p* = 0.109; n-${\mathrm{MIRHS}}_{high-}$: $b_{1}$ = −0.00665, *p* = 0.674). In contrast, higher values of $\beta_{H}$ predicted significantly larger n-${\mathrm{MIRHS}}_{low}$ ($b_{1}$ = 0.464, *p* = 4.37 × 10^−308^). The decompositions based on reward feedback revealed that this effect was strongest for n-${\mathrm{MIRHS}}_{low-}$, which was significantly higher for larger values of $\beta_{H}$ (standardized coefficient: $b_{1}$ = 0.523, *p* = 1.53 × 10^−253^; dashed lines in Figure 5f, right). In comparison, n-${\mathrm{MIRHS}}_{low+}$ was also significantly predicted by $\beta_{H}$, but to a lesser degree ($b_{1}$ = 0.121, *p* = 1.90 × 10^−19^; solid lines in Figure 5f, right). These results suggest that the influence of reward harvest rate on promoting heuristic win–stay/lose–switch is primarily mediated by increased information value of no reward in the low-reward state. That is, the larger $\beta_{H}$, corresponding to the effect of reward harvest rate, was associated with less negative n-MIRHS_*low*−_ toward zero, thus increasing the informativeness of receiving no reward during this state. This can be used to detect modulations of choice behavior by reward harvest rate as these effects can change across blocks of trials.

### 3.4. Revealing the Presence of Alternative Learning Strategies in Multidimensional Reward Environments

Finally, we examined whether the information-theoretic metrics can be utilized to identify the presence of alternative learning strategies in multidimensional reward environments. In naturalistic settings, reward outcomes often depend on distinct features or attributes of the available choice options. To mimic such scenarios, we simulated a task in which only one of the two choice attributes—either the shape or the color of the stimuli—was associated with reward, thereby introducing uncertainty about which attribute was predictive of reward outcomes (Figure 6a; see Section 2.3.4 for more details). We considered three RL agents, each exhibiting distinct types of choice strategy: (1) an agent that only learns and chooses based on the color attribute (*Color-only*), (2) an agent that equally weighs the color and shape attributes without arbitration (*No Arbitration*), and (3) an agent which dynamically arbitrates between two attributes based on the reliability of each attribute in predicting reward (*Dynamic Arbitration*; Figure 6b).

Examining the arbitration weight ω across different block types revealed that the *Dynamic Arbitration* agent successfully adapted its strategy to the environment; it consistently increased ω during *Color-to-Color* blocks, indicating a growing reliance on color (magenta curve in Figure 6e, top), and shifted ω toward a shape-based strategy following reversals in *Color-to-Shape* blocks (magenta curve in Figure 6f, top). When reversals occurred within the color attribute (*Color-to-Color* blocks), the performance trajectories (P(Better)) were qualitatively similar across all agents, as expected. In each model, performance dropped immediately after reversals followed by gradual recovery (Figure 6e, bottom). As expected, the *Color-only* agent reached its peak performance more quickly (Figure 6e, bottom), as it did not consider shape values, which were irrelevant in this condition. During *Color-to-Shape* blocks, when the color information was no longer predictive of rewards after reversals, the *Dynamic Arbitration* agent performed the best (Figure 6f, bottom), as this agent appropriately identified the correct reward-predictive attribute by dynamically arbitrating between the two feature dimensions (Figure 6f, bottom).

Although the learning strategy of the *Color-only* agent can be readily distinguished by its P(Better) trajectory during the *Color-to-Shape* blocks, distinguishing among all three agents is more challenging in *Color-to-Color* blocks based on performance alone. Therefore, we examined whether information-theoretic metrics could be used to distinguish among the three types of learning strategies represented by the RL agents. Although the underlying arbitration weight ω can, in principle, be estimated through model fitting, behavioral metrics offer a more direct window into learning and decision-making strategies. Moreover, these metrics can be computed over a subset of trials and require far simpler computations compared to model fitting, which depends on continuity in the choice data.

To quantify reward-dependent strategies, we computed the conditional entropy of reward-dependent strategies (ERDS) separately for each attribute—${\mathrm{ERDS}}_{Shape}$ and ${\mathrm{ERDS}}_{Color}$ (Figure 7a,b). We found that during *Color-to-Color* blocks, the entropy of the shape-based strategy (${\mathrm{ERDS}}_{Shape}$) was close to its maximum value of 1 for all the three agents, reflecting the fact that shape carried no information about reward in this block type (dashed lines in Figure 7a). In contrast, the entropy of the color-based strategy (${\mathrm{ERDS}}_{Color}$) exhibited modulation across trials, with distinct dynamics for each agent (solid lines in Figure 7a). Consistent with the performance trajectories, the *Color-only* agent showed the lowest entropy, followed by the *Dynamic Arbitration* agent (rank-sum test on ${\mathrm{ERDS}}_{Color}$, *p* = 7.98 × 10^−34^). To assess the relative dominance of the two learning strategies, we computed the difference between the two entropy values, ${\mathrm{ERDS}}_{Shape}-{\mathrm{ERDS}}_{Color}$. The resulting positive values indicated that all three agents correctly prioritized the color-based learning strategy during this block type, as reflected by lower entropy for the relevant dimension (i.e., color) (Figure 7c). These two metrics thus provide insight beyond the trajectories of arbitration weight ω (Figure 6e top), which does not directly capture changes in the learned values (Q) that could drive choice behavior.

During the *Color-to-Shape* blocks, the entropy values (${\mathrm{ERDS}}_{Shape}$) indicated near-random use of the shape-based strategy for all three agents prior to the reversal (dashed lines in Figure 7b). After the reversal—when the shape attribute became informative of reward—the *Dynamic Arbitration* showed the most pronounced adjustment, characterized by a decreased reliance on the color-based strategy (reflected by a rise in ${\mathrm{ERDS}}_{Color}$; solid magenta line in Figure 7b), and an increased reliance on the shape-based strategy (reflected by a drop in ${\mathrm{ERDS}}_{Shape}$; dashed magenta line). The difference in the two entropy values further revealed that only the agents capable of learning both attributes (*No Arbitration* and *Dynamic Arbitration*) were able to shift their strategies after the reversal, as indicated by ${\mathrm{ERDS}}_{Color}>{\mathrm{ERDS}}_{Shape}$ (Figure 7d). In contrast, the *Color-only* agent exhibited ${\mathrm{ERDS}}_{Color}<{\mathrm{ERDS}}_{Shape}$ throughout the entire block.

To quantify potential interactions between the color-based and shape-based learning strategies, we measured the correlation between ${\mathrm{ERDS}}_{Shape}$ and ${\mathrm{ERDS}}_{Color}$, computed separately from two distinct periods within each block. “Non-stationary” trials were defined as the first 30 trials at the beginning of each block and those immediately following the reversal (highlighted by red bars in Figure 7a,b). In contrast, “steady” trials corresponded to the final 10 trials before the reversal or at end of each block, during which performance had plateaued (blue bars in Figure 7a–d).

This analysis revealed distinct patterns of interaction between the two learning strategies across the three agents (Figure 7e,f). More specifically, the *Color-only* agent showed no significant correlation between ${\mathrm{ERDS}}_{Shape}$ and ${\mathrm{ERDS}}_{Color}$ during either periods of the *Color-to-Shape* blocks (non-stationary: *r* = 0.137, *p* = 0.174; steady: *r* = 0.072, *p* = 0.477; Figure 7e). This result is expected, as the *Color-only* agent does not employ any shape-based strategy, which renders the response to reward based on shape effectively random. In contrast, the *No Arbitration* agent exhibited competitive interaction between the two strategies, as indicated by a significant negative correlation between ${\mathrm{ERDS}}_{Shape}$ and ${\mathrm{ERDS}}_{Color}$ (non-stationary: *r* = −0.435, *p* = 6.08 × 10^−6^; steady: *r* = −0.386, *p* = 7.43 × 10^−5^; Figure 7e). This pattern reflects the agent’s decision-making strategy, in which color and shape information are always weighted equally, regardless of which attribute is currently predictive of reward (fixed ω=0.5). Interestingly, the *Dynamic Arbitration* agent exhibited different trend of interaction across the two periods of the block. During the non-stationary phase, this model exhibited a significant negative correlation between ${\mathrm{ERDS}}_{Shape}$ and ${\mathrm{ERDS}}_{Color}$ (*r* = −0.374, *p* = 1.28 × 10^−4^). However, this correlation disappeared during the steady phase of the block (*r* = −0.0409, *p* = 0.686), when the color-based strategy became dominant (i.e., ω>0.5). This shift in correlation pattern can be used to detect dynamic arbitration between alternative learning strategies.

We found overall consistent results during the *Color-to-Shape* blocks (Figure 7f). The *Color-only* agent showed no significant interaction between strategies in either period (non-stationary: *r* = −0.0346, *p* = 0.733; steady: *r* = −0.055, *p* = 0.587). In contrast, the *No Arbitration* agent exhibited significant negative correlations during both phases of the block (*r* = −0.522, *p* = 2.59 × 10^−8^; steady: *r* = −0.269, *p* = 0.00685). Notably, only the *Dynamic Arbitration* agent—capable of adapting its behavior based on the relative reliability of the two attributes in predicting reward—exhibited a shift in the interaction pattern over time. Specifically, it showed a significant negative correlation during the non-stationary period (*r* = −0.536, *p* = 9.19 × 10^−9^), which weakened and became non-significant during the steady phase (*r* = −0.151, *p* = 0.135), consistent with the pattern observed in the *Color-to-Color* blocks. Together, these results demonstrate that a negative correlation between ${\mathrm{ERDS}}_{Shape}$ and ${\mathrm{ERDS}}_{Color}$ during the non-stationary phase and the absence of such a correlation during the steady phase is indicative of dynamic arbitration between alternative learning strategies.

## 4. Discussion

In this study, we demonstrate how behavioral metrics inspired by information theory can be used to identify certain learning and decision-making mechanisms. In particular, we applied metrics based on conditional entropy, normalized mutual information, and their decompositions based on specific outcomes to choice behavior during two variants of the probabilistic reversal learning task—a widely adopted paradigm for studying cognitive flexibility across species. Using these metrics, we investigated whether specific neural or computational mechanisms—specified by the reinforcement learning (RL) models serving as ground truth—could be inferred from choice behavior. To that end, we examined positivity bias, gradual changes in the learning rates due to reward-dependent metaplasticity, the influence of reward harvest rate, and the adoption and arbitration between alternative learning strategies in multidimensional environments.

One of the key strengths of the proposed information-theoretic metrics lies in their versatility and flexibility. In contrast, fitting RL models to choice behavior in order to identify underlying mechanisms requires a continuous stream of choice and reward data, with the precision of the estimated parameters heavily dependent on the amount of available data. As our results demonstrate, information-theoretic metrics circumvent these limitations, as they can be computed from as few as several dozen trials, even when drawn from non-contiguous segments of the data. Directly comparing the information content of our metrics with regression weights for predicting choice (Figure 3j), we found similar decoding performance from both types of measures. However, information-theoretic metrics offer the distinct advantage of being simple and model-agnostic, whereas regression-based features are inherently model-dependent—requiring decisions about which predictors to include, how many lags to consider, how to evaluate model fit and statistical significance, and how to interpret the resulting regression weights. Future work is needed for a more principled comparison between information-theoretic and regression-based approaches, for example, to identify the conditions under which each performs best. Nonetheless, the two approaches can work in tandem to examine behavior.

Moreover, time courses of information-theoretic metrics can be computed for each trial point by concatenating trial-wise data across different blocks. This approach serves as a useful visualization tool for examining behavioral changes over time. Finally, the analysis in Figure 7 demonstrates how metrics computed from different periods within a block can provide insight into temporal changes in learning and choice strategies. Taken together, the efficiency and flexibility of these metrics make them valuable model-agnostic tools for identifying and testing hypotheses about the mechanisms underlying learning and decision making, in a way that complements the more traditional model-dependent methods.

Another important advantage of information-theoretic metrics is their adaptability to target specific variables of interest within a given study. As illustrated in simulations of reward harvest rate effects on behavior (Figure 5), the key variable—reward harvest state—can be incorporated into the mutual information metric n-MIRHS to assess how reward harvest rate influences choice behavior. This measure revealed the role of reward harvest rate in shaping choice strategy under varying reward contingencies by quantifying the information flow from prior reward outcomes and reward states into the agent’s choice strategy. Therefore, these metrics can be flexibly formulated to match the demands of a given research context and provide quantifiable insights into the role of specific variables.

In turn, the insights provided by these metrics can guide the development of improved models that better capture key aspects of behavioral data [32]. As numerous studies have pointed out [80,81,82], the best-fitting model among a set of candidates may still fail to reproduce important features of choice behavior. Therefore, validating candidate models through posterior predictive checks remains a critical step in modeling [29,83]. Information-theoretic metrics offer ideal tools for this process, as they provide summary statistics specific to the variables of interest. For example, different RL models could be validated by comparing information-theoretic metrics computed from simulated data with those derived from empirical data [32,40]. By quantifying the divergence between empirical observations and model-generated data, this comparison both reveals a model’s limitations and pinpoints the mechanisms that must be added for improvement [32]. Such an approach is especially valuable when standard model-comparison metrics are inconclusive—for example, with small or noisy datasets that inherently favor simpler, low-parameter models. In those cases, one can use information-theoretic measures to constrain the candidate models by selecting the one whose simulated statistics most closely match the empirical data. Lastly, although this study focuses on RL models, the same framework can be applied to other model classes—such as those based on Bayesian inference [35,84].

In addition to serving as useful tools for model discovery and validation, information-theoretic metrics may also contribute to data-driven approaches in studies involving choice behavior. Our decoding analyses in Section 3.1 and Section 3.2 serve as proof-of-concept, showing that the information-theoretic measures contain sufficient information to detect positivity bias when the ground truths are known. This example highlights the broader potential of the metrics in detecting latent cognitive states or mechanisms other than positivity bias (which may lack ground truths in real data). For example, one could train a decoder to predict experimentally defined task states from either behavioral measures (reaction time, reward rate, pupil dilation) or neural recordings (spike trains, EEG, fMRI) by using information-theoretic features extracted from choice behavior [85]. In this approach, the experimentally defined states serve as the ground-truth labels during training, and the decoder’s ability to recover those labels is then evaluated on held-out behavioral (or neural) data. Future research could apply this model-agnostic yet interpretable framework to infer latent cognitive or neural states.

There are some challenges and considerable potential for extending the information-theoretic framework presented here. For example, while our entropy measures quantify the overall consistency in the stay/switch strategy in response to reward and other relevant variables, they do not directly capture directionality—that is, they do not indicate what causes what. As a result, these metrics should be considered alongside other measures to provide a more complete picture. This limitation makes interpreting mutual information versus conditional entropy particularly challenging. Moreover, while information-theoretic metrics capture sensitivity to reward feedback, they cannot be uniquely mapped onto the learning rates. Finally, although we have focused on conditional entropy and normalized mutual information, other behavioral metrics based on other concepts from information theory can be developed, including transfer entropy [86,87], mutual information between discrete and continuous variables [88,89,90], and partial information decomposition [91,92], among others. Incorporating these extensions would allow the behavioral metrics to be generalized to more complex tasks, including those involving more than two alternative options, higher-dimensional feature spaces, and task contexts.

## Figures and Tables

**Figure 1 entropy-27-01056-f001:**
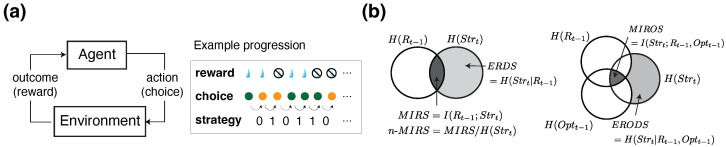
Probabilistic reversal learning task and information-theoretic framework for describing agent-environment interaction. (**a**) Illustration of agent–environment interaction and an example progression of task-related information used for computing behavioral information-theoretic metrics. (**b**) Left: Illustration of conditional entropy and mutual information between previous reward ($R_{t-1}$) and current strategy ($Str_{t}$). Gray area represents conditional entropy of reward-dependent strategy (ERDS), and black area represents mutual information between strategy and previous reward (MIRS). n-MIRS denotes mutual information normalized by H(Str). Right: Similar metric defined by the joint combination of previous reward ($R_{t-1}$) and chosen option ($Opt_{t-1}$). Gray area represents conditional entropy of reward and option-dependent strategy (ERODS), and black area represents mutual information between reward outcome, choice options, and choice strategy (MIROS).

**Figure 2 entropy-27-01056-f002:**
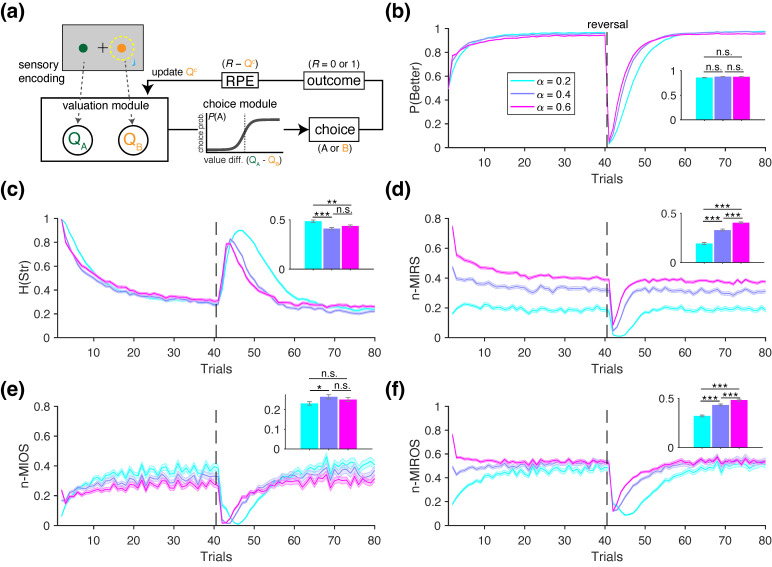
Schematic of an RL agent performing a probabilistic reversal learning task, with behavioral quantification using information-theoretic metrics. (**a**) Schematic of a standard RL algorithm that makes choices based on the value difference between the two options ($Q_{A}-Q_{B}$), using a softmax function to compute choice probabilities (P(A)). After each choice, the model updates the value estimate of the chosen option, $Q^{c}$, based on the difference between the reward outcome (*R*) and $Q^{c}$. This difference is known as the reward prediction error (RPE). (**b**) Plot shows average performance over time, defined as probability of choosing the better-rewarding option, shown separately for three different learning rates: α=0.2,0.4,0.6. Inset shows the average performance over 100 simulated blocks. ‘n.s.’ denotes no significant difference. (**c**–**f**) Plots show the averaged time course of the entropy of strategy (H(Str), (**c**)); normalized mutual information between strategy and previous reward (n-MIRS, (**d**)); normalized mutual information between strategy and option (n-MIOS, (**e**)); and normalized mutual information between strategy and the previous reward/option combinations (n-MIROS, (**f**)). Asterisks indicate significant differences based on a two-sided rank-sum test (*: *p* < 0.05; **: *p* < 0.01; ***: *p* < 0.001).

**Figure 3 entropy-27-01056-f003:**
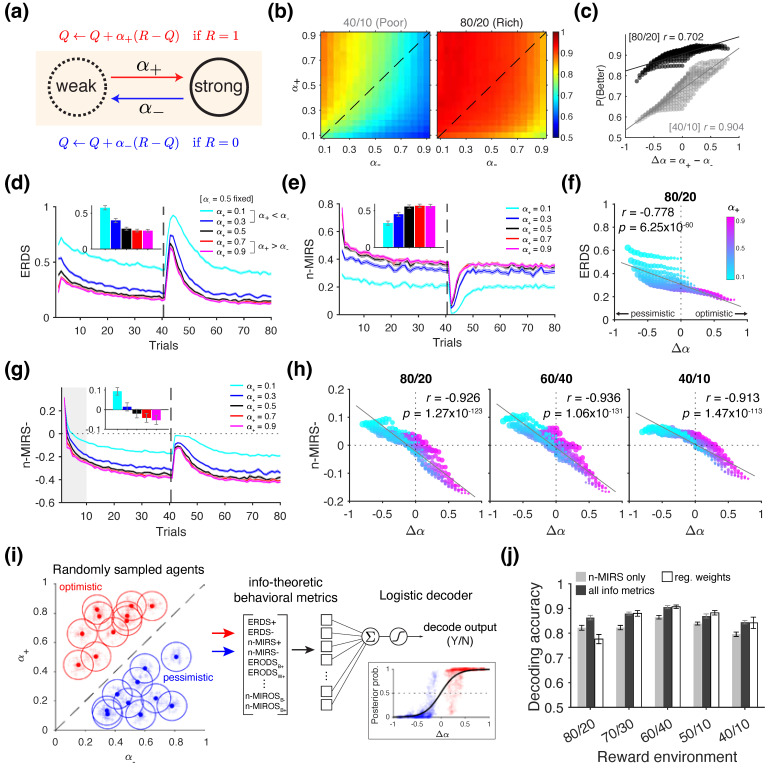
Behavioral signatures of positivity bias and its decoding using information-theoretic metrics. (**a**) Illustration of value updates following rewarded (R=1) and unrewarded (R=0) outcomes, with corresponding differential learning rates $\alpha_{+}$ and $\alpha_{-}$, respectively. (**b**) Behavioral signature of positivity bias. Heatmaps show the average performance for each combination of ($\alpha_{+}$, $\alpha_{-}$) in two example reward schedules. (**c**) Correlations between performance and the difference in the learning rates ($\Delta \alpha =\alpha_{+}-\alpha_{-}$), shown separately for two reward schedules. (**d**,**e**) Example time courses of ERDS (**d**) and n-MIRS (**e**) for five example RL agents with different learning rates. Red/magenta lines indicate “optimistic” agents, blue/cyan lines indicates “pessimistic” agents, and black corresponds to the neutral agent. Insets show metric computed over the whole block. (**f**) Correlation between Δα and ERDS metric in the 80/20% environment. (**g**) Example time courses for the decomposition of n-MIRS after negative reward feedback (n-MIRS-). The sign of n-MIRS-, computed from the first 10 trials (inset), is predictive of positivity bias. (**h**) Correlations between Δα and n-MIRS- metric, shown separately for three example reward environments. (**i**) Illustration of sampling and decoding procedure to predict the sign of Δα using information-theoretic metrics. For each RL agent in the two groups—”optimistic” with Δα>0 or “pessimistic” with Δα<0)—a mean value of ($\alpha_{+}$, $\alpha_{-}$) was randomly drawn. Learning rates for each session were then randomly sampled from Gaussian distributions centered on these means. Behavioral metrics were computed from the resulting choice behavior in each session and used to train a binary decoder. The inset on the right shows the predicted posterior probability generated by cross-validated decoders for each true value of Δα, with a sigmoid curve fitted to the data. (**j**) Cross-validated decoding accuracy from decoders trained on the information-theoretic metrics—n-MIRS metric only (gray) or all available metrics (black)—and logistic regression weights (white empty bars), separately for different reward environments.

**Figure 4 entropy-27-01056-f004:**
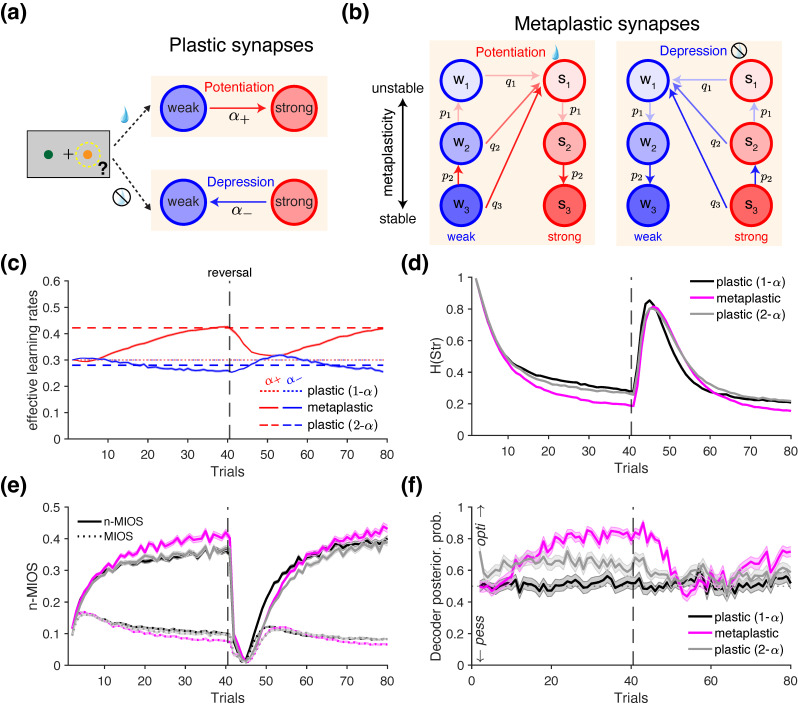
Comparison of plastic and metaplastic models and their distinct behavior revealed by information-theoretic metrics. (**a**) Schematic of a biological implementation of a standard RL model with binary synapses that undergo stochastic reward-dependent plasticity. In this model, “weak” synapses transition to the “strong” synaptic state (i.e., are potentiated) with probability $\alpha_{+}$ following reward, whereas “strong” synapses transition to the “weak” state (i.e., are depressed) in the absence of reward. (**b**) Schematic of synapses with reward-dependent metaplasticity, consisting of three meta-states with increasing levels of stability (i=1,2,3), each associated with one of the two levels of synaptic strengths (“weak” *w* or “strong” *s*). Transition probabilities $q_{i}$ govern changes from the *i*^th^ meta-state to the most unstable meta-state of opposite efficacy, with $q_{1}>q_{2}>q_{3}$. Transition probabilities $p_{i}$ govern transitions between meta-states of the same efficacy, and are also higher for less stable states (i.e., $p_{1}>p_{2}$). (**c**) Averaged trajectories of “effective learning rates” for the metaplastic model (solid lines), a plastic model with a single learning rate (dotted lines; α=0.3); the plastic model with differential learning rates is estimated from the choice behavior of the metaplastic model (dashed lines). Red and blue correspond to the effective learning rates on rewarded and unrewarded trials, respectively. (**d**) Averaged trajectory of H(Str), shown separately for the three models as indicated in the legend. (**e**) Averaged trajectory of normalized mutual information between previously chosen option and strategy (n-MIOS, solid lines), separately for each of the three model. MIOS before normalization by H(Str) are plotted for reference (dotted lines). (**f**) Posterior probability that a given set of metrics was classified as originating from an “optimistic” (*opti*) vs. a “pessimistic” (*pess*) agent, based on binary decoders applied to the choice behavior of the three models.

**Figure 5 entropy-27-01056-f005:**
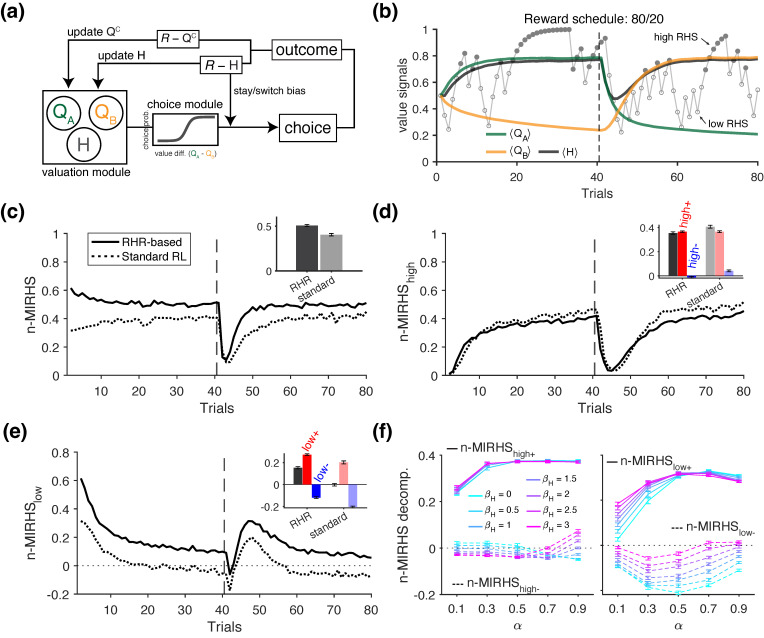
Detecting the modulation of choice strategy by reward harvest rate. (**a**) Schematic of an RL model that tracks reward harvest rate to modulate choice strategy. In this model, the difference between the immediate reward (*R*) and reward harvest rate (*H*) is used to directly influence win–stay and lose–switch behavior. (**b**) Averaged time course of the model estimated values ($Q_{i}$’s in green and orange) and reward harvest rate (*H* in black) during the PRL task with a reversal on trial 40. The gray trace in the background depicts the time course of *H* from an example session. Filled (empty) circles represent trials with high (low) reward harvest states (RHS), determined via a median split of *H* values across the session. Simulation parameters: α=0.3, β=10, $\beta_{H}=2$. (**c**) Averaged time course of normalized mutual information between strategy and reward/RHS combinations (n-MIRHS), shown for models with and without the reward harvest rate modulation. (**d**) Decomposition of the n-MIRHS metric for trials in the high RHS state. Inset shows further decompositions into previously rewarded (*high+*, red) and unrewarded (*high−*, blue) trials. (**e**) Decomposition of the n-MIRHS metric for trials in the low-RHS state. Inset shows further decompositions into previously rewarded (*low+*, red) and unrewarded (*low−*, blue) trials. (**f**) Plotted are the mean simulated n-MIRHS decompositions based on high (left panel) or low (right panel) reward state, following rewarded (solid) or unrewarded (dashed lines) trials, as a function of α (X-axis) and $\beta_{H}$ (indicated by colors). The influence of $\beta_{H}$ was most pronounced in the information gained from no reward in the low RHS.

**Figure 6 entropy-27-01056-f006:**
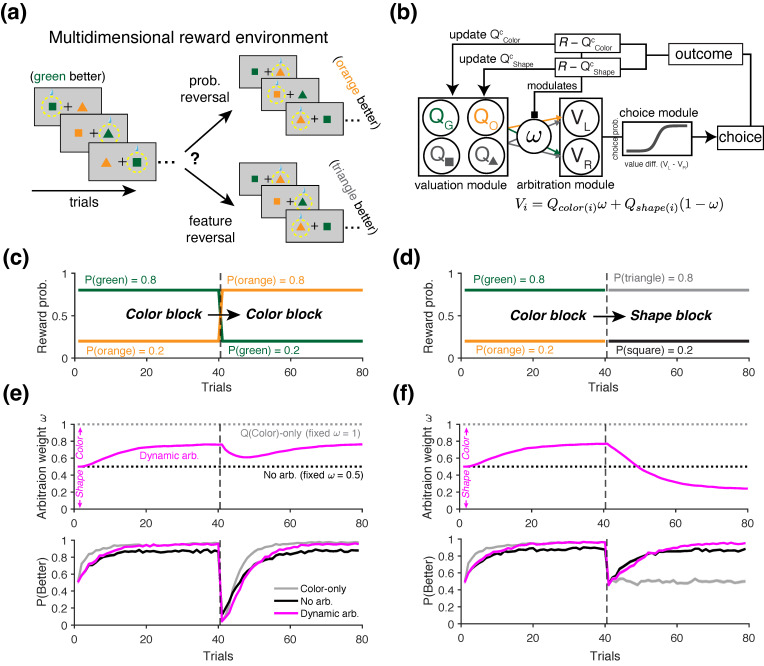
Schematic of a multidimensional reward learning task and RL models used to perform it. (**a**) Illustration of example trials from the multidimensional reversal learning task. Reversals can occur either in the reward probabilities associated with two options (e.g., switching from green to orange as the better option), as in the standard probabilistic reversal learning task, or in the feature dimension predictive of rewards (e.g., from color to shape). (**b**) Schematic of the dynamic arbitration model that simultaneously tracks the values of individual color and shape features (*Q*). These feature values are then combined to compute the overall value (*V*) of the left and right options. The arbitration weight (ω) is dynamically adjusted based on the reliability of the chosen color and shape values, estimated from their respective reward prediction errors. (**c**) Illustration of a reversal where the better option reverses from the green to the orange object (Color-to-Color reversal), regardless of shape. (**d**) Illustration of a reversal where the better option reverses from the green to triangle object (Color-to-Shape, feature reversal). (**e**) Averaged time courses of arbitration weight (ω, top) and task performance (bottom) during Color-to-Color reversal blocks for the three models. The dynamic arbitration model (*Dynamic arb.*) gradually increases its weighting of color. The model with no dynamic arbitration (*No arb.*) maintains a fixed weight of ω=0.5, equally combining color and shape values. The *Color-only* model does not track shape values and therefore, ω remains fixed at 1. (**f**) Same plots as in (**e**) but for Color-to-Shape (feature reversal) blocks.

**Figure 7 entropy-27-01056-f007:**
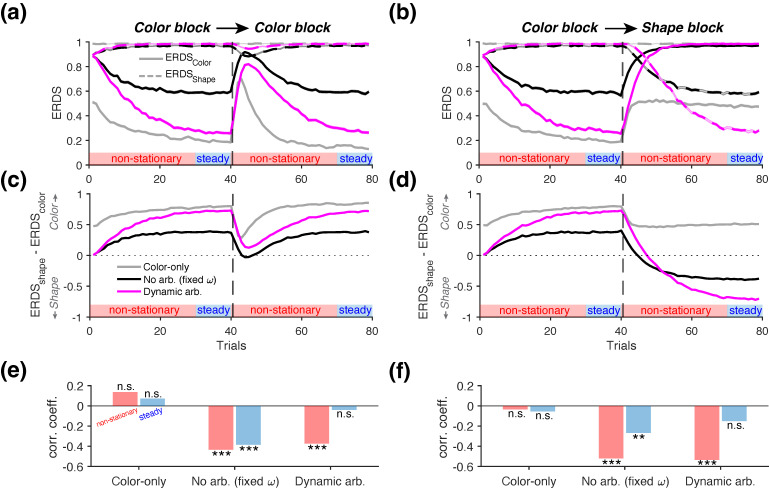
Distinct predictions of information-theoretic metrics for adopting and arbitrating between alternative strategies in a multidimensional reward environment. (**a**) Time courses of the conditional entropy of reward-dependent strategy based on color (${\mathrm{ERDS}}_{Color}=H\left(Str_{Color}|R\right)$, solid lines) and shape (${\mathrm{ERDS}}_{Shape}=H\left(Str_{Shape}|R\right)$, dashed lines) during the Color-to-Color blocks. Line colors indicate the three simulated agents (gray: *Color-only*; black: *No Arbitration*; magenta: *Dynamic Arbitration*). (**b**) Similar to (**a**) but for the Color-to-Shape (feature reversal) blocks. (**c**) Time course of the difference ${\mathrm{ERDS}}_{Shape}-{\mathrm{ERDS}}_{Color}$ during the Color-to-Color blocks, measuring the relative dominance of two learning strategies. Higher values indicate dominance of the color-based strategy. (**d**) Similar to (**c**) but for the Color-to-Shape (feature reversal) blocks. (**e**,**f**) Correlation coefficients between ERDS*_Color_* and ERDS*_Shape_* during the Color-to-Color (**e**) and the Color-to-Shape (feature reversal) blocks (**f**), computed separately from the “non-stationary” (first 30 trials of each block) or “steady” portion of the block (last 10 trials before the reversal or end of the block). Asterisks indicate significance level (**: p<0.01; ***: p<0.001; n.s.: not significant). The *Color-only* agent showed no interaction between the two strategies, with no significant correlations observed between ERDS*_Color_* and ERDS*_Shape_* in any block or phase. The *No Arbitration* agent (with fixed ω = 0.5) exhibited strong negative interaction between the two strategies, as reflected by significant negative correlations during both steady and non-stationary portions of the blocks. The *Dynamic Arbitration* agent showed significant negative interaction during the non-stationary phase, but this correlation became non-significant during the steady phase as one strategy began to dominate.

## Data Availability

The analysis codes used for this study are available at https://github.com/DartmouthCCNL/woo_etal_entropy (accessed on 22 May 2025).

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
