# Peer review of "An Information-Theoretic Framework for Understanding Learning and Choice Under Uncertainty"

_entropy, 2025, doi:10.3390/e27101056_

Round 1

Reviewer 1 Report

Comments and Suggestions for Authors

Woo and collaborators present an information-theoretic framework to analyze binary choices in reversal tasks. The information measures are very informative and can distinguish between model classes, which seems to be an important achievement. The paper is easy to read and there are many good examples to illustrate the main points.

One problematic aspect that I see is that some of the main conclusions come from decoders that, using the information measures, can distinguish between models, but in real data we will not know what the ground truth. In addition, if the model classes used to generate the results are not representative of the real data, then the main conclusion might differ. What aspects of the results are assumed to be robust to big changes of the model? I think that a discussion on this would be important.

In addition, the author claims several times that behavioral measures can be sometimes more important than model comparison to draw conclusions about the underlying mechanisms (e.g., positivity bias being dependent on the difference in learning rates after correct or wrong responses). Although I generally agree with this idea, the current paper does not have a comparison between the two approaches. I presume that if that comparison were made, it should be possible to draw equivalent conclusions and being able to distinguish between models with the same amount of data required to compute the behavioral measures. At least, the author can make a deeper discussion on when model comparison will fail as compared to behavioral measures, as in the current paper the behavioral measures are also calibrated based on some model assumptions.

The global reward rate in Eq. 43 is an internal variable. Therefore, it seems that conditioning to it in Eq. 46 might not be useful in real data. It will be only useful if that internal variable can be estimated, but this would require optimizing alpha_G. In other words, using this internal for conditioning requires model fitting, and then it is unclear why the authors do not go all the way to fit the full model to estimate alpha+ and alpha-. Maybe explaining the rationale of this hybrid approach would be good.

What is the rationale for using the soft-max function in Eq. 37? References can be provided. Indeed, references to

https://www.pnas.org/doi/abs/10.1073/pnas.0710743106

https://www.nature.com/articles/s41467-024-49711-1

where it is shown that soft-max functions naturally appear under rationality assumptions.

The reliability difference in Eq. 54s seems to weight reliable options, but in general the most reliable option does not need to be the more promising and worth exploring to maximize reward rate.

There are several Bayesian models in the literature on reversal learning, e.g.,

https://www.pnas.org/doi/epdf/10.1073/pnas.1615773114

A comparison or discussion about them seems to be in order.

Minor:

-The title should reflect that the framework is for discrete choice

-Notation wise, it might be convenient to indicate delays in variables. For instance, in eq. 6 the variable Str refers to current trial, and Opt to previous trial, but when looking a the equations, one might not remember that the conditional is on the previous trial.

-In Fig 5c, the global reward rate might not coincide with winning or losing, but maybe there is a correlation between them.

Comments on the Quality of English Language

Good quality

Reviewer 2 Report

Comments and Suggestions for Authors

Woo et al. approach the problem of analyzing behavioral data in learning and decision-making tasks through information theory metrics lens. Given the extensive use of information theory in quantifying correlations between neural responses and external factors in other areas of neuroscience, a systematic investigation of their potential application in choice behavior seems relevant. However, we believe the current manuscript can significantly improve by following suggestions:

  • On page 15 authors mention that:

“Overall, this example illustrates that information-theoretic metrics are sensitive to the underlying mechanisms as subtle as the learning rate—differences that may be obscured  when relying solely on summary measures such as performance”.

 Given the already widespread use of summary statistics and regression methods in analyzing behavioral experiments in the field, it seems important to actually verify this claim in the results. For example, in the first experiment, how well does the positivity bias decoder perform if trained and tested on the summary statistics of or even raw choice data instead of information-theory metrics? In general, it seems like this and all the following simulations can significantly improve in clarity if authors provide some comparison with logistic and linear regression analysis ( e.g. probability of strategy as dependent and previous choice and reward as independent variables).

  • The authors mention that:

“…information-theoretic metrics … can be computed from as few as several dozen trials, even when drawn from non-contiguous segments of the data.”

However, their method uses entropy of strategy H(Str) presumably from the empirical distribution of stay/switch choices. This might not be an issue in their setting where they can simulate choice behavior over 10,000 blocks but it seems like the practical utility of this method for human/animal behavior is contingent on having an appropriate empirical distribution. Is this true? Could the author showcase the feasibility of their method in a previously published human behavior dataset, one with no access to unlimited simulations and where the behavioral strategy is analyzed with other methods?

  • In the methods section, the authors extensively explain the MIROS metric for calculating mutual information of joint distribution of previous option and reward and strategy. This metric looks a little problematic since the two variables, reward and previous option, are correlated and it would not be clear how the mutual information between their joint and strategy would be interpreted (either one the variables could be informative or their combination could be informative, basically there’s no way to decompose that from the joint MI). However, upon reading the result section, it doesn’t even seem like this metric was utilized except on page 15. ( I thought I spotted it on the formulae on page 13 too, but it looks like that’s a typo?). Could authors provide some justification for why they introduced the mutual information of reward/choice and strategy and how they plan to interpret this metric (compared to individual MI between option/reward and strategy)?

  • The manuscript seems to have a lot of typos (e.g. Line 508 Page 17, sign of r seems to be flipped?). I suggest a careful proofreading before re-submission.

Round 2

Reviewer 1 Report

Comments and Suggestions for Authors

The authors have fully addresed my comments. 

Author Response

We thank the reviewer for the feedback and are happy to hear that all of their comments have been addressed. 

Reviewer 2 Report

Comments and Suggestions for Authors

The authors have revised the manuscript to address the majority of our previous concerns. However, we still have one remaining concern related to a mismatch between claims and the evidence used to support them. Based on the data provided, it seems that both logistic regression and information-theoretical metrics are useful methodologies for analyzing data. In specific instances one might have advantages over the other but such advantages are application specific. The authors do show that in certain situations information theory metrics are more informative that logistic regression coefficients. However, they don’t provide any principled way of determining in what situations such methods would have benefits beyond more standard logistic regression-based techniques. Thus, it seems like this is a major limitation that should be clarified – that information theory based tools explored here are but one tool in the potential arsenal and that before applying them it is not easy to know whether they will work better or worse than more standard approaches.

Furthermore, the authors claim that the main advantage of information-theoretic metrics is that they are model-free. First, as a minor point, we would suggest using “model agnostic” rather than “model-free” since the latter terminology has come to take a very different meaning in the RL community. But more generally, it is just not clear why having a model-agnostic approach is better. Are there situations where the assumptions in regression would impede discovery? Or, do the assumptions just differ across the methods in a manner that is unrelated to their utility or performance? Again, it would be useful if the authors could clarify so that claims more clearly align with the presented evidence.

Author Response

Comment 1: The authors have revised the manuscript to address the majority of our previous concerns. However, we still have one remaining concern related to a mismatch between claims and the evidence used to support them. Based on the data provided, it seems that both logistic regression and information-theoretical metrics are useful methodologies for analyzing data. In specific instances one might have advantages over the other but such advantages are application specific. The authors do show that in certain situations information theory metrics are more informative that logistic regression coefficients. However, they don’t provide any principled way of determining in what situations such methods would have benefits beyond more standard logistic regression-based techniques. Thus, it seems like this is a major limitation that should be clarified – that information theory based tools explored here are but one tool in the potential arsenal and that before applying them it is not easy to know whether they will work better or worse than more standard approaches.

Response 1: We thank the reviewer for the feedback and are happy to hear that most of their comments have been addressed. 

We also agree with the reviewer that the evidence presented in our study does not directly support the superior utility of one method over the other in all cases. Indeed, this would require another study using a more principled way of comparison between them. For this reason, we have mainly highlighted the efficiency, flexibility, and interpretability of the information-theoretical metrics, in a way that complements the model-dependent methods (such as regression or reinforcement learning models) rather than arguing for the superiority of one over the other. To emphasize this point, we have added the following sentence in Discussion (Line #775). Please see our next comment below related to this point.

Future work is needed for a more principled comparison between information-theoretic and regression-based approaches, for example, to identify the conditions under which each performs best. Nonetheless, the two approaches can work in tandem to examine behavior.

Comment 2: Furthermore, the authors claim that the main advantage of information-theoretic metrics is that they are model-free. First, as a minor point, we would suggest using “model agnostic” rather than “model-free” since the latter terminology has come to take a very different meaning in the RL community. But more generally, it is just not clear why having a model-agnostic approach is better. Are there situations where the assumptions in regression would impede discovery? Or, do the assumptions just differ across the methods in a manner that is unrelated to their utility or performance? Again, it would be useful if the authors could clarify so that claims more clearly align with the presented evidence.

Response 2: We thank the reviewer for this comment about model-free connotation in RL. We agree with the reviewer and have now changed the wording into “model-agnostic” throughout the manuscript. 

Regarding the general point, we think that the main advantage of being model-agnostic is its convenience and simplicity in the interpretation. For example, suppose that one has observed a low decoding accuracy by using methods based on a (model-dependent) regression coefficient. In this case, the researcher is introduced to additional uncertainty about what has caused the poor performance of the decoder—it could be attributed to the poor quality of fit in the regression model itself, or to the true underlying lack of information in the data. Hence, one cannot conclude with confidence until testing different regression models with different parameters. We believe that information-theoretic measures allow one to skip this intermediary step by being model-agnostic. However, we do agree with the reviewer that these assumptions (model-agnostic vs. model-dependent) themselves are not directly related to their performance, as we have also observed similar performance between two methods. Rather, we think that the contribution of our measures lies in providing a more efficient model-agnostic alternative that complements the model-dependent methods, instead of competing with them.

To clarify this point further, we have now revised the sentence in Discussion (Line #782) as follows:

Taken together, the efficiency and flexibility of these metrics make them valuable model-agnostic tools for identifying and testing hypotheses about the mechanisms underlying learning and decision making, in a way that complements the more traditional model-dependent methods.